There are amendments to this paper

# Glucose homeostasis is regulated by pancreatic β-cell cilia via endosomal EphA-processing

Francesco Volta[1,2], M. Julia Scerbo[1], Anett Seelig[1], Robert Wagner[3,4,5], Nils O'Brien [1], Felicia Gerst[3,4,5], Andreas Fritsche[3,4,5], Hans-Ulrich Häring[3,4,5], Anja Zeigerer [5,6], Susanne Ullrich[3,4,5] & Jantje M. Gerdes [1,5]*

*Diabetes mellitus* affects one in eleven adults worldwide. Most suffer from Type 2 *Diabetes* which features elevated blood glucose levels and an inability to adequately secrete or respond to insulin. Insulin producing β-cells have primary cilia which are implicated in the regulation of glucose metabolism, insulin signaling and secretion. To better understand how β-cell cilia affect glucose handling, we ablate cilia from mature β-cells by deleting key cilia component *Ift88*. Here we report that glucose homeostasis and insulin secretion deteriorate over 12 weeks post-induction. Cilia/basal body components are required to suppress spontaneous auto-activation of EphA3 and hyper-phosphorylation of EphA receptors inhibits insulin secretion. In β-cells, loss of cilia/basal body function leads to polarity defects and epithelial-to-mesenchymal transition. Defective insulin secretion from IFT88-depleted human islets and elevated pEPHA3 in islets from diabetic donors both point to a role for cilia/basal body proteins in human glucose homeostasis.

[1] Institute for Diabetes and Regeneration Research, Helmholtz Zentrum München, Ingolstädter Landstr. 1, 85764 Neuherberg, Germany. [2] Technical University of Munich, Medical Faculty, 81675 Munich, Germany. [3] Department of Internal Medicine IV, Division of Endocrinology, Diabetology, Nephrology, Vascular Disease and Clinical Chemistry, University Hospital of Tübingen, Tübingen, Germany. [4] Institute for Diabetes Research and Metabolic Diseases of the Helmholtz Centre Munich at the University of Tübingen (IDM), Tübingen, Germany. [5] Deutsches Zentrum für Diabetesforschung, DZD, Munich, Germany. [6] Institute for Diabetes and Cancer, Helmholtz Zentrum München, Munich, Germany. *email: jantje.gerdes@helmholtz-muenchen.de

Primary cilia are antenna-like organelles that protrude from the plasma membrane of many different cell types. The basal body anchors the microtubular ciliary axoneme to the cell and serves as an organizing structure for protein complexes involved in protein trafficking to and from the primary cilium. Protein trafficking is mediated by Intra-Flagellar Transport proteins (IFTs) that act as adaptors between cargo and motor proteins kinesin-II for anterograde and cytoplasmic dynein for retrograde transport. Although primary cilia were observed as early as the late 19th century, their biological role only experiences renewed appreciation with the identification of a class of diseases with impaired cilia function, called ciliopathies[1]. Primary cilia are found in a number of metabolically active tissues, including cardiomyocytes, hypothalamic neurons, and pancreatic islets. Two ciliopathies, Bardet-Biedl syndrome (BBS) and Alström Syndrom, are characterized by increased appetite and food consumption resulting in truncal obesity. Diabetes is more common in Alström and BBS patients compared with obese controls[2].

BBS is an oligogenic disease that is linked to more than 20 disease genes. While the most common loci are *BBS1* and *BBS10* (OMIM #209900 and #615987, respectively), *BBS4* is linked to severe obesity[3]. Importantly, *Bbs4*$^{-/-}$ mice have impaired glucose handling and blunted first-phase insulin secretion as a prelude to obesity. Insulin receptor (IR) is recruited to the primary cilia of stimulated β-cells in these mice, and downstream signaling nodes are not activated in islets with impaired basal bodies or cilia[4].

β-cells secrete insulin and are therefore important regulators of glucose metabolism. In addition to the canonical pathway of insulin secretion involving ATP-production, membrane depolarization and subsequent opening of voltage-gated $Ca^{2+}$-channels, ultimately leading to insulin secretion, there are other signaling pathways modulating insulin secretion[5]. The subclass of Ephrin-type A receptors/Ephrin-type A (EphA/EphrinA) are implicated as regulators of insulin secretion[6]. Eph receptors are the largest known family of receptor protein–tyrosine kinases, and Ephrins and their receptors are juxtacrine signaling components. Under basal conditions, levels of phosphorylated EphA increase in β-cells. EphA phosphorylation inhibits Rac family small GTPase 1 (Rac1) activity and suppresses insulin secretion. Increased glucose concentration recruits more Ephrin ligand to the cellular surface and changes downstream processing of the signal, facilitating insulin release[6].

Over the past years, a connection between ciliary signaling pathways and endosomal trafficking is emerging. Ciliogenesis requires vesicle docking to the mother centriole of an elongated centrosome in many cell types, including fibroblasts and smooth muscle[7,8]. In *Trypanosome brucei*, endocytic and exocytotic events are coordinated at the ciliary pocket, which serves as the only site for endo- and exocytosis[9]. Signaling pathways form a highly complex network of feedback loops and other regulatory mechanisms. An important element of this network is endosomal processing[10], which serves to suppress spontaneously activated unoccupied receptor protein–tyrosine kinases in addition to ligand-activated endocytosis. Here, we report on the role of primary cilia in β-cell function by ablating cilia and glucose homeostasis. Removal of ciliary/basal body components leads to aberrant cell polarity and EMT impairing actin dynamics. As a result, spontaneously activated EphA3 is no longer internalized efficiently to be dephosphorylated. Elevated pEphA3 levels block insulin secretion acutely thus leading to defective insulin secretion and glucose intolerance.

## Results

**β-cell cilia are required for adult glucose homeostasis.** To investigate whether glucose metabolism is regulated by primary

cilia in adult β-cells, we generated a β-cell-specific inducible cilia knockout that we abbreviate as βICKO mice (pronounced Bicko). We started with a floxed mouse strain of the *Ift88* gene that, if deleted, ablates primary cilia[11]. We then crossed these mice with β-cell-specific *Pdx1-CreER* mice carrying a transgene placing the tamoxifen-inducible *cyclic recombinase-estrogen receptor* (*Cre-ER*) fusion construct under the transcriptional control of the *pancreatic and duodenal homeobox 1* promoter region[12]. We induced *Ift88* gene knockout by Tamoxifen (Tx)-administration at 4 weeks of age and followed glucose tolerance over a total of 12 weeks (Fig. 1a; Supplementary Fig. 1a). To control for effects of Tx-treatment and *Pdx1-CreER* overexpression, both vehicle-treated βICKO mice and Tx-treated *Ift88*$^{loxP/loxP}$ mice from the starter strain served as controls. Efficiency of recombination was assessed on the genomic DNA levels as well as by quantification of cilia in isolated pancreatic islets, and both were reduced by 80% or more (Supplementary Fig. 1b, c). We followed the cohort of induced βICKO animals and controls over time. Glucose handling was significantly impaired in the Tx-treated βICKO animals at 4 weeks (Supplementary Fig. 2a (repeated measures one-way ANOVA); area under the curve (AUC) (veh) = 778 mg dL$^{-1}$ glucose ± 75 (s.e.m.); AUC (Tx) = 1091 mg dL$^{-1}$ ± 105 (s.e.m.); one-way ANOVA, Holm-Sidak multiple comparison).

After 8 weeks, glucose handling is significantly impaired at $t = 90$ min (Supplementary Fig. 2b; repeated measures one-way ANOVA) and total AUC is more than twofold increased (AUC (veh) = 682 mg dL$^{-1}$ ± 114, AUC (Tx) = 1536 mg dL$^{-1}$ ± 212, one-way ANOVA, Holm-Sidak multiple comparison).

Glucose intolerance worsened to pronounced glucose intolerance with blood glucose values significantly higher than vehicle-treated controls at 12 weeks post induction (Fig. 1a; $t = 60$ vehicle = 309 mg dL$^{-1}$ ± 37 Tx: 540 ± 36; $t = 90$ vehicle:237 mg dL$^{-1}$ ± 27 Tx:530 mg dL$^{-1}$ ± 40; $t = 120$ vehicle:193 mg dL$^{-1}$ ± 17 Tx:477 mg dL$^{-1}$ ± 48 (repeated measures one-way ANOVA)). The AUC of blood glucose expenditure increased almost 2.5-fold (AUC (veh) = 714 ± 164 mg dL$^{-1}$ (s.e.m.), AUC (Tx) = 1704 ± 169 mg dL$^{-1}$, one-way ANOVA, Holm-Sidak multiple comparison). This suggests that β-cell primary cilia are essential for glucose homeostasis. To account for possible side effects of Tamoxifen treatment or *Pdx1-CreER* expression, we included two different control groups, βICKO mice treated with oil and *Ift88*$^{loxP/loxP}$ mice treated with tamoxifen. Glucose tolerance was not affected in both animals, and there were no statistically significant differences between the two controls groups (Supplementary Fig. 2f. (repeated measures one-way ANOVA)).

In parallel to glucose testing, we also determined in vivo insulin secretion in response to stimulation with 2 g/kg intraperitoneal glucose at 8 and 12 weeks post induction, and observed significantly blunted acute insulin secretion in Tx-treated animals at both time points (Fig. 1b; group comparison (repeated measures one-way ANOVA) Supplementary Fig. 2c; group comparison (repeated measures one-way ANOVA)). Overall, these results show that β-cell cilia are required for adult glucose homeostasis and β-cell function.

**Ift88 is required for β-cell survival.** Attenuated insulin secretion can be caused by loss of β-cells and/or by β-cell failure to respond. At 6 weeks post induction, 2 weeks after the first manifestation of glucose intolerance, β-cell mass did not significantly differ between Tx-treated and control animals. Therefore, loss of β-cell cilia leads to impaired insulin secretion that is independent of β-cell mass (Fig. 1c). After 20 weeks, β-cell mass is lowered approximately sixfold in Tx-treated animals compared with controls (Fig. 1d; $n = 3$; 5886 μm$^2$ × (mm$^2$)$^{-1}$ ± 709 vs 963 μm$^2$ × mm$^{-2}$ ± 62).

A previous study of *bbs* gene knockdowns in zebrafish described increased proliferation in β-cells and higher rates of

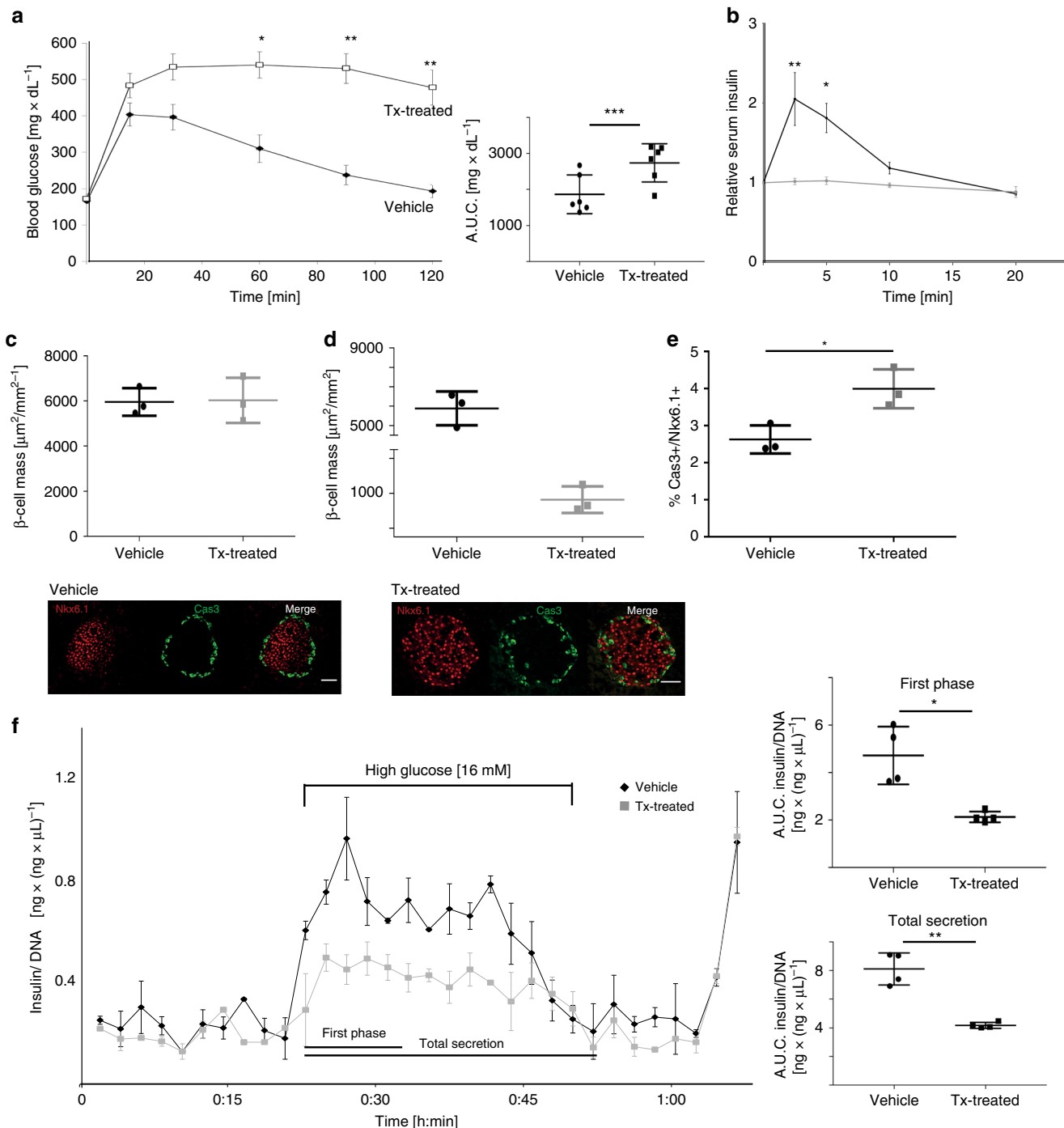

**Fig. 1 Characterization of βICKO mice phenotype. a** Intraperitoneal glucose tolerance test at 12 week post induction ($n = 6$, $p < 0.0001$ (GTT, repeated measures one-way ANOVA); $p = 0.002$ (AUC, one-way ANOVA, Holm-Sidak multiple comparison) mean ± s.e.m.). **b** Insulin-secretion test. ($n = 3$, $p < 0.0001$ (GTT, repeated measures one-way ANOVA), mean ± s.e.m.) 12 weeks post induction. **c** β-cell mass 6 weeks post induction ($n = 3$, mean ± s.d.). **d** β-cell mass 20 weeks post induction ($n = 3$, $p = 0.0006$ (t test), mean ± s.d.). **e** Percentage of apoptotic beta cells over total of beta cells. Representative images of control and treated animal islets. Nkx6.1 shown in red, and caspase-3 in green ($n = 3$ animals, $n = 10$ islet per animal, 20 weeks post induction, scale bar 50 μm, $p = 0.0216$, mean ± s.d.). **f** Dynamic insulin secretion of isolated βICKO islets (Tx-treated shown in gray, veh shown in black). AUC quantification of first phase (time 18:00 to 30:00 min) and total secretion (from 18:00 to 48:00 min) (First phase $p = 0.0058$; total secretion $p = 0.0005$ (t test), islets pooled from $n = 10$ animals, experiment repeated $n = 4$, and insulin measured $n = 2$ per experiment, mean ± s.d.

apoptosis when exposed to high glucose concentrations[13]. We therefore tested β-cell proliferation and apoptosis, by Ki-67 and Caspase-3 immunofluorescence, respectively, but found no change at 6 weeks post induction, in our model (Supplementary Fig. 2d, e). Twenty weeks post induction, however, there was higher apoptosis in β-cells of Tx-treated βICKO mice compared

with controls (Fig. 1e). Higher apoptosis rates could explain the loss of β-cells over time and thus implicate Ift88 and cilia function in β-cell survival.

This suggests that we have a two-pronged phenotype: the acute effect is on β-cell function and the long-term effect leads to β-cell loss.

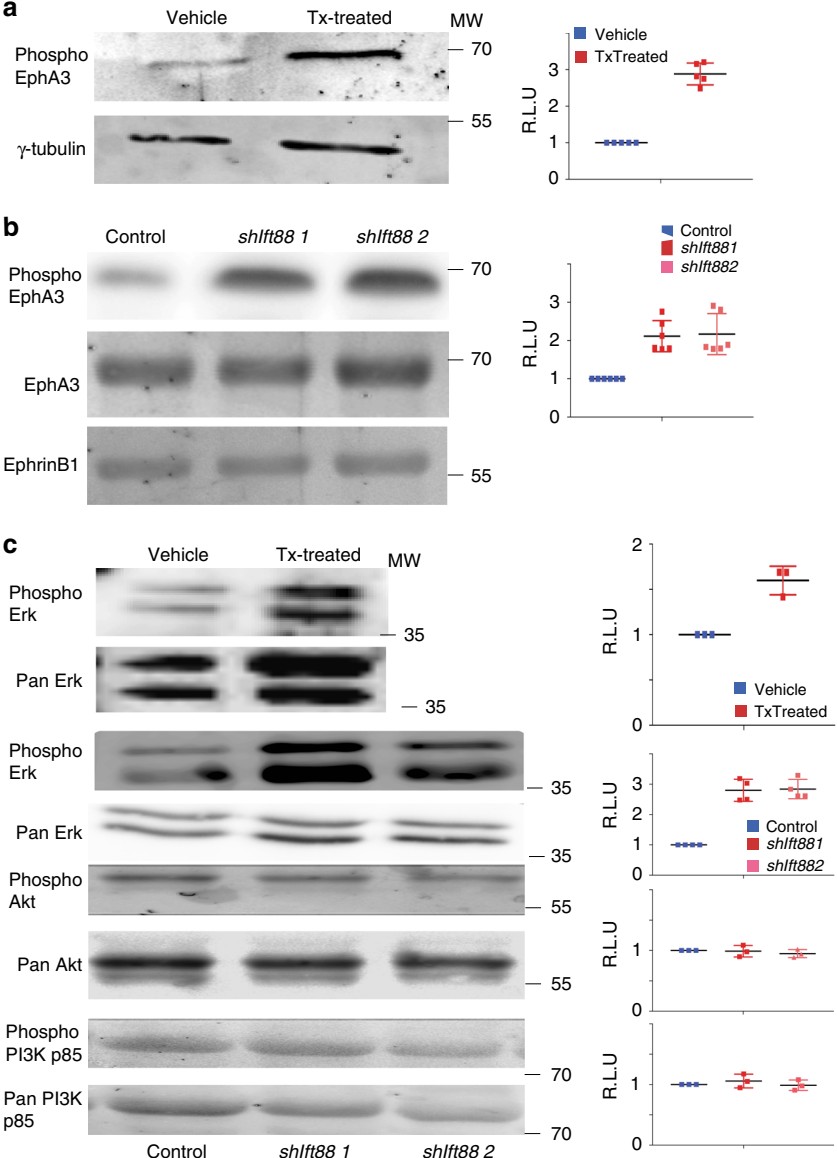

**Fig. 2 EphA3 hyperphosphorylation in *Ift88*-depleted β-cells. a** pEphA3 (Tyr779) and γ-tubulin protein levels in Tx- and veh-treated islets and quantification (*n* = 5, mean ± s.d.) **b** pEphA3 (Tyr779), EphA3, and EphrinB1 protein levels in shIft88 1 and 2 and scrambled RNA (control) and quantification. (*n* = 6, mean ± s.d.) **c** Phosphorylated p44/42 ERK/MAPK (Thr202/Tyr204) and ERK/MAPK protein levels (*n* = 4) in primary islets and MIN6m9 cells, pAkt (Ser437), Akt (*n* = 3), phosphorylated p85 (Tyr458) and p85 protein levels (*n* = 3) in MIN6m9 cells and quantification, mean ± s.d.

**Ex vivo insulin secretion is impaired in βICKO islets**. In adult mice, *Pdx1* is almost exclusively expressed in mature β-cells but *Pdx1-Cre* also effectively induces recombination in the hypothalamus, suggesting *Pdx1* is expressed in the brain during development[14].

To test if the observed effects on glucose metabolism are the result of β-cell or hypothalamic cilia dysfunction, we took advantage of the fact that primary islets lose innervation during the isolation process. We thus isolated βICKO islets to test glucose-stimulated insulin secretion ex vivo. Islets from Tx-treated βICKO mice 20 weeks post induction had significantly reduced insulin secretion after incubation with 11 mM glucose for 30 min (Supplementary Fig. 3a, vehicle:7.1 ± 1.2; Tx:4.6 ± 0.5). Consistently, islets from 14-week-old male *Bardet-Biedl Syndrome 4* (*Bbs4⁻/⁻*) knockout mice also had lower insulin secretion in the presence of glucose (Supplementary Fig. 3b).

For further functional and mechanistic studies, we isolated βICKO islets and treated them in vitro with1 μM Tx overnight to

remove *Ift88* and impair cilia function[15]. In Tx-treated islets, both phases of insulin secretion were reduced 5d after treatment (Fig. 1f; first phase Tx = 4.7 ± 1, veh:2.2 ± 0.2; *p* = 0.0058; total secretion Tx:8.1 ± 0.97, veh:4.2 ± 0.18 (*t* test). Because these effects were observed in isolated, de-innervated islets, we conclude that the effect of ciliary ablation is a direct effect of β-cell cilia, and not mediated via neuronal input.

To further exclude an involvement of primary cilia in the hypothalamus, we tested for hypothalamic recombination in *ROSA^{mT/mG}* reporter mice crossed to the inducible *Pdx1-CreER* mouse line that we used to generate the βICKO mouse line[12,18]. Under our chosen conditions, we did not detect any sign of recombinatory activity in the third ventricular region of the hypothalamic brain (Supplementary Fig. 3c). This demonstrates that the βICKO mouse phenotype is pancreas specific (Supplementary Fig. 3d). Based on these findings, we can rule out an involvement of hypothalamic cilia in the βICKO phenotype. To test for β-cell specificity of *Pdx1-CreER*-driven recombination, we

examined glucagon- and insulin-expressing islet cells in pancreatic sections from these reporter mice (Supplementary Fig. 3e). We did not observe recombination in glucagon-positive cells of these mice, indicating that at the time of induction, *Pdx1* is specific to β-cells.

**EphA2/3 is hyperphosphorylated in βICKO islets**. Primary cilia are not only involved in insulin signaling, and we suspected that at least one other signaling pathway regulating insulin is implicated in the βICKO phenotype, because it is more severe than the effect of β-cell-specific Insulin Receptor knockout mice, of which only 25% develop severe glucose intolerance and diabetes[4,16,17]. To better understand which other pathway(s) might be misregulated, we screened 27 RTK for differential activation in Tx-treated βICKO and *Bbs4*[−/−] islets compared with controls under standard culture conditions (Supplementary Fig. 4a, b). Out of 27 candidates, five were upregulated by over 50% in Tx-treated βICKO islets: the proto-oncogenes Met and Ret, FMS-like tyrosine kinase 3, along with EphA2 and 3. Three non-receptor tyrosine kinases were also >50% upregulated. These included S6 ribosomal protein, a known downstream effector of mTor signaling, which is an important ciliary signaling pathway[18] and finally p44/p42 MAPK and Stat1, two downstream effectors of Eph/Ephrin signaling[19,20]. Ephrin-type A receptors 2 and 3 (EphA3 and EphA2) were among the most misregulated in both Tx-treated βICKO and *Bbs4*[−/−] islets. We chose to focus on EphA receptors, as EphA/EphrinA signaling is an important regulator of insulin secretion[6]. The finding could in part explain the (early) effects of ciliary impairment in Tx-treated βICKO islets. We confirmed EphA3 hyperphosphorylation in Tx-treated βICKO islets by immunoblotting (Fig. 2a; Tx-treated islets = 288% ± 26 compared with control). We were unable to validate excessive levels of pEphA2 due to a lack of suitable antibodies. Because EphA2/3 is hyperphosphorylated in islets from two different knockout models, it is likely directly linked to ciliary/basal body function.

**EphA3 hyperphosphorylation activates ERK/MAPK signaling**. To better understand the molecular underpinnings of EphA2/A3 upregulation, we next tested if depletion of *Ift88* mRNA resulted in similar EphA3 upregulation in an insulinoma cell line, Min6m9[21]. We generated two clonal cell lines stably expressing short hairpin RNA targeting the 3′ UTR of *Ift88* mRNA (shIft88 1/2). In these cell lines, Ift88 is reduced 75% at the mRNA and 80% at the protein level; ciliation was reduced 85% (Supplementary Fig. S1d–f). pEphA3 levels increased more than twofold compared with controls (Fig. 2b). At the same time, total protein levels of EphA3 did not change. Protein levels of EphrinB1 were also unchanged in shIft88 Min6m9 cells (Fig. 2b).

Hyperphosphorylation of a receptor does not necessarily implicate overactivation of the downstream signaling pathway. EphA-downstream signaling nodes include phosphatidylinositol-3-kinase (PI3K) and Akt on the one hand and extracellular regulated MAPkinase/mitogen-activated-protein-kinase (ERK/MAPK) on the other. In Tx-treated βICKO islets, we observed an increase of ERK/MAPK phosphoactivation (Fig. 2c). Similarly, we observed more than twofold increased ERK/MAPK phosphorylation in both shIft88-expressing MIN6m9 cell lines compared with control (Fig. 2c; n = 4). Phosphorylation of the p85 PI3K-regulatory subunit or Akt was unchanged in the two cell lines (Fig. 2c; n = 3). In conclusion, loss of *Ift88* and concomitantly increased EphA3 phosphorylation leads to aberrant activation of the ERK/MAPK signaling pathway.

**Aberrant EphA activation blocks insulin secretion**. EphA/Ephrin A signaling regulates insulin secretion from murine islets both in vivo and ex vivo[6]. Therefore, increased EphA2/3 phosphorylation could explain decreased insulin secretion from βICKO islets challenged with glucose. If misregulated EphA2/3 signaling is the underlying mechanism of the observed insulin-secretory phenotype, we should be able to exacerbate or ameliorate the defect by manipulating the ratio of pEphA3 to EphA3. To test our hypothesis, we perfused Tx- and vehicle-treated βICKO islets with 1 µg ml[−1] recombinant EphrinA5-Fc that has been shown to bind all EphA receptors[22]. Subsequently, we tested dynamic insulin secretion 1 h after EphrinA5-Fc addition (Fig. 3a).

When perfused with 11 mM glucose and 1 µg ml[−1] EphrinA5-Fc, insulin secretion was slightly but not significantly elevated in vehicle-treated βICKO islets (control). By contrast, EphrinA5-Fc treatment rescued glucose-stimulated insulin secretion in Tx-treated βICKO islets. Islets depleted of *Ift88* and subsequently treated with EphrinA5-Fc secreted similar levels of insulin to control islets under both low and high glucose conditions (Fig. 3a). This strongly suggests that EphA2/3-misregulation inhibits insulin secretion in Tx-treated βICKO islets. In addition, this reversal suggests that β-cells with impaired ciliary function can still respond to EphrinA, albeit at higher concentrations.

To further substantiate the involvement of misregulated EphA/EphrinA5-signaling in the ciliary insulin-secretion defect, we treated *Ift88*-depleted MIN6m9 cells with 1 µg ml[−1] EphrinA5-Fc and determined the stimulation index, calculated by normalizing the amount of insulin secreted in response to a glucose stimulus to baseline. We found that treatment restored glucose-stimulated insulin secretion (Fig. 3b; control = 1.805 ± 0.09 shIft88 = 1.803 ± 0.085). Moreover, we transiently transfected shIft88 MIN6m9 cells with a dominant negative isoform of EphA5 (DN-EphA5) that is lacking the cytoplasmic domain and therefore does not propagate the EphA signaling cascade[6]. DN-EphA5 overexpressing shIft88 MIN6m9 cells increase insulin secretion similar to control cells when stimulated with 16.7 mM glucose (Fig. 3c; control = 1.805 ± 0.092; shIft88 = 1.578 ± 0.038). Insulin secretion was similar to that of control cells, and basal insulin secretion was elevated in control cells transfected with DN-EphA5. Consequently, the stimulation index decreased, further corroborating the link between ciliary dysfunction, EphA signaling, and insulin secretion (Supplementary Fig. 5a). As expected, *Ift88* mRNA suppression results in a near-complete loss of glucose-responsiveness when testing glucose-stimulated insulin secretion (Fig. 3b; control = 1.805 ± 0.09; shIft88 = 1.05 ± 0.053). Phenotypic overlap between *Bbs4*[−/−] and Tx-treated βICKO islets and shIft88 expressing MIN6m9 cells suggests that blunted insulin secretion is a direct result of Ift88 depletion and ciliary dysfunction. To further rule out potential off-target effects, we transfected cells from both clonal shIft88 lines with murine *Ift88* cDNA and tested glucose-stimulated insulin secretion. Replenishing *Ift88* restores glucose-stimulated insulin secretion to levels similar to mock-transfected control cells, confirming that the cellular phenotype is linked to loss of *Ift88* function (Fig. 3b; control = 1.868 ± 0.02; shIft88 + Ift88 = 1.807 ± 0.073).

To further probe the mechanism by which EphrinA5-Fc treatment restores glucose-stimulated insulin secretion, we tested if and how EphA3 tyrosine phosphorylation changes under the experimental conditions. Increased EphrinA5 levels reduce pEphA levels in the context of high glucose[6]. At the same time, acute addition of recombinant EphrinA5-Fc results in increased EphA phosphorylation. In Tx-treated and control βICKO islets, EphA3 phosphorylation increases 5 min after EphrinA5-Fc addition (Supplementary Fig. 5a)[6]. In contrast, we observed a

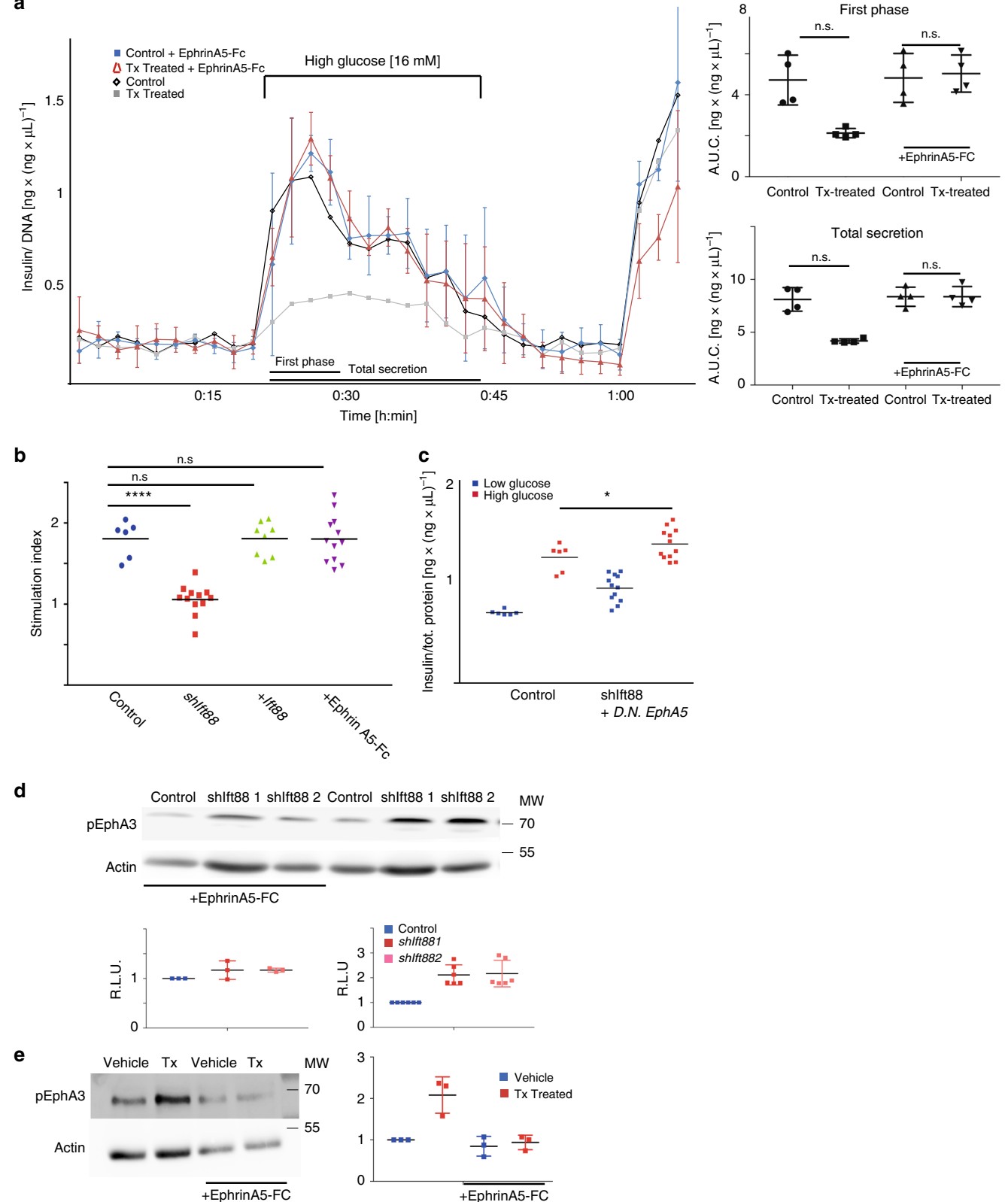

reduction in pEphA3 in the presence or absence of *Ift88* mRNA, 1 h after EphrinA5-Fc addition (Fig. 3d; Supplementary Fig. 5a).

The relative difference of pEphA3 levels in both EphrinA5-Fc-treated *shIft88*-expressing clonal cell lines was reduced compared with basal conditions (Fig. 3d; shIft88 1 = 211% compared with control, shIft88 2 = 216% compared with control (*n* = 6); *shIft88* +

EphrinA5-Fc = 120% compared with control, *shIft882* + EphrinA5-Fc = 117% compared with control (*n* = 6)).

Similarly, we found reduced pEphA3 levels in Tx-treated βICKO islets 1 h after EphrinA5-Fc stimulation (Fig. 3e). This observation suggests that EphA3 hyperphosphorylation blocks insulin secretion and that EphrinA5-Fc treatment restores insulin

**Fig. 3 The insulin secretion phenotype is linked to EphA/ EphrinA signaling. a** Dynamic insulin secretion of isolated βICKO islets treated with tamoxifen (blue) or vehicle (Ethanol, red) stimulated with 1 μg ml$^{-1}$ EphrinA5-Fc. On the right, quantification of first phase (AUC; time 18:00 to 30:00 min) and total secretion (AUC from 18:00 to 48:00 min) (islets pooled from $n = 10$ animals, experiment repeated $n = 4$ and insulin measured n = 2 per experiment, mean ± s.d.). In gray, Tx-treated βICKO islets and in black vehicle-treated βICKO islets from Fig. 1f for comparison. **b** Insulin secretion of *shIft88* expressing MIN6m9 cells ($n = 6$ control, 12 shIft88 experiments, insulin measured $n = 2$ per experiment; $p < 0.0001$ (t test)); *shIft88* MIN6m9 cells transfected with murine *Ift88* (2 μg plasmid per well) ($n = 4$ control, eight shIft88 experiments, insulin measured $n = 2$ per experiment); shIft88 cells stimulated with 1 μg ml$^{-1}$ EphrinA5-Fc ($n = 6$ control, 12 shIft88 experiments, insulin measured $n = 2$ per experiment). **c** Insulin secretion of shIft88 cells and shIft88 cells transfected with dominant negative DN-EphA5 (1 μg plasmid per well) ($n = 6$ control, 12 shIft88 experiments, insulin measured $n = 2$ per experiment; $p = 0.017$ (t test)). **d** pEphA3 (Tyr779) and actin levels in shIft88 1 and 2) and scrambled RNA (control) treated with 1 μg ml$^{-1}$ ephrinA5-Fc and quantification ($n = 3$, mean ± s.d.). **e** pEphA3 (Tyr779) and actin protein levels in Tx- and vehicle-treated islets treated with 1 μg ml$^{-1}$ EphrinA5-Fc and quantification ($n = 3$, mean ± s.d.).

secretion by dephosphorylating EphA3 in Ift88-depleted MIN6m9 cells. In summary, manipulations of several components of the EphA/EphrinA signaling cascade restore insulin secretion in primary pancreatic islets or MIN6m9 cells depleted of *Ift88*. These lines of evidence strongly support the conclusion that misregulation of the EphA/EphrinA signaling cascade is at the base of aberrant insulin secretion in our Tx-treated βICKO mice. In addition, EphA/EphrinA signaling is an important effector of insulin secretion.

**Ciliary impairment blocks endosomal recycling.** What is the cause of EphA hyperphosphorylation in βICKO islets? Increased signaling could be achieved by either enhancing (auto-) kinase or blocking phosphatase activity. We found no change in EphA3 protein levels or Ephrin ligands in absence of Ift88 (Fig. 2). It seems that cilia function does not antagonize EphA-forward signaling. Consequently, we interrogated the negative regulation of EphA phosphorylation.

It has long been demonstrated that receptor-protein-tyrosine-kinases (RTK) auto-phosphorylate spontaneously[23]. To counteract spontaneous signaling activity, activated RTKs including EphA are continuously transported to the phosphatase-rich perinuclear region[24–26]. Endocytosis is initiated by ligands binding to their receptors. Following internalization, vesicles are transported to early endosomes where cargo is either returned to the plasma membrane directly, transported to perinuclear recycling endosomes, or degraded via late endosomes and lysosomes[10]. We hypothesized that EphA internalization is perturbed when cilia/basal body function is interrupted.

To better understand if and at which stage EphA3 internalization is perturbed, we used a modified pulse-chase surface biotinylation protocol. Surface proteins on *Ift88*-depleted cells were labeled with biotin and precipitated with neutravidin beads (Fig. 4a, $t = 0$ min). There were no differences for EphA3, although there was more surface-bound pEphA3 in *Ift88*-depleted cells (Fig. 4a). After surface biotinylation, we added EphrinA5-Fc and incubated the cells at 37 °C for 60 min. The cell surface was then stripped to remove all biotin. Neutravidin precipitation will select proteins that were on the cell surface but have been internalized. After 60 min, pEphA3 was 60% reduced in *Ift88*-depleted cells, indicative of defective endocytosis (Fig. 4b). To determine the amount of receptor recycled back to the cell surface, we incubated cells at 37 °C for 60 min before another round of biotin stripping (Fig. 4c). At this time, biotin labels from cells that have been recycled to the cell surface are removed. In *Ift88*-depleted cells, levels of biotinylated EphA3 and pEphA3 were both ~twofold upregulated (Fig. 4c), suggesting that recycling to the plasma membrane is perturbed. Because EphA3 is dephosphorylated in the perinuclear endosomal compartment and we see a rise in pEphA3 levels, we conclude that EphA3 recycling is stalled before pEphA3 cargo reaches the perinuclear endosomal recycling compartment.

To obtain additional evidence that cilia are involved in endocytic uptake of EphrinA5, we treated *shIft88* cells with biotin-labeled EphrinA5-Fc over 75 min (Fig. 4d). After fixation, cells were incubated with fluorescently labeled streptavidin to detect and quantify biotinylated EphrinA5-Fc[27]. Control cells internalize EphrinA5-Fc with a delay of ~15 min and start to saturate at 45 min (Fig. 4d). By contrast, *shIft88* cells had a strong reduction in EphrinA5-Fc internalization marked by approximately threefold lower saturation plateau and a 2.5-fold reduced internalization rate. This indicates that EphrinA5-Fc occupies fewer endosomal compartments. Altogether, these data suggest an impairment of endosomal transport processes in *Ift88*-depleted cells.

Next, we tested if EphrinA5-Fc uptake was also impaired in βICKO islets. EphrinA5-Fc-related fluorescence intensity in Tx-treated islets was significantly lower than in vehicle-treated controls after 60 min (Fig. 4e). Because EphrinA5 binds to EphA receptors, this suggests that ligand-stimulated EphA internalization is impaired in *Ift88*-depleted β-cells and that this leads to an accumulation of EphA on the plasma membrane. This accumulation prevents proper transport of EphA3 to the perinuclear region and consequently results in failure to dephosphorylate EphA3.

None of the commercially available antibodies raised against EphA3 were suitable for immunofluorescent staining in our hands. Therefore, we used a myc-tagged EphA3 expression plasmid, *EphA3-myc*, to visualize EphA3 localization (Fig. 4f). After stimulation with EphrinA5-Fc, EphA3-myc is efficiently internalized and predominantly found in the perinuclear region of control cells. By contrast, a large subpopulation of EphA3-myc is observed in the periphery of *Ift88*-depleted cells, proximal to the plasma membrane. Taken together these data suggest that EphA3 internalization depends on Ift88 function.

One of the most important negative regulators of EphA-mediated signaling activity is protein-tyrosine-phosphatase-1B (Ptp1b) encoded by *Ptpn1* (protein-tyrosine-phosphatase, non-receptor type 1; ref. [25]). Global loss of Ptp1b activity in *Ift88*-deficient cells seems unlikely because, unlike βICKO mice, *Ptpn1*$^{-/-}$ mice have improved glucose tolerance and insulin sensitivity[28]. Ptp1b localizes to the perinuclear region, and it has been shown that pEphA is internalized and transported to the perinuclear region to be dephosphorylated by Ptp1b[24,25].

Ift88 loss could lead to impaired endocytosis, thus effectively blocking Ptp1b de-phosphorylation. To test this, we treated *Ift88*-depleted cells with a global protein-tyrosine-phosphatase (PTP) inhibitor (PanPtp). As expected, pEphA3 levels increased significantly upon treatment. Importantly, the difference in pEphA3 levels between control and *shIft88* MIN6m9 cells was completely abolished when all PTPs were inhibited (Fig. 5a). A Ptp1b-specific inhibitor also abolished the difference in pEphA3 (Fig. 5a). Interestingly, phosphorylation levels were higher when all PTPs were inhibited (PanPtp), suggesting that Ptp1b is not the

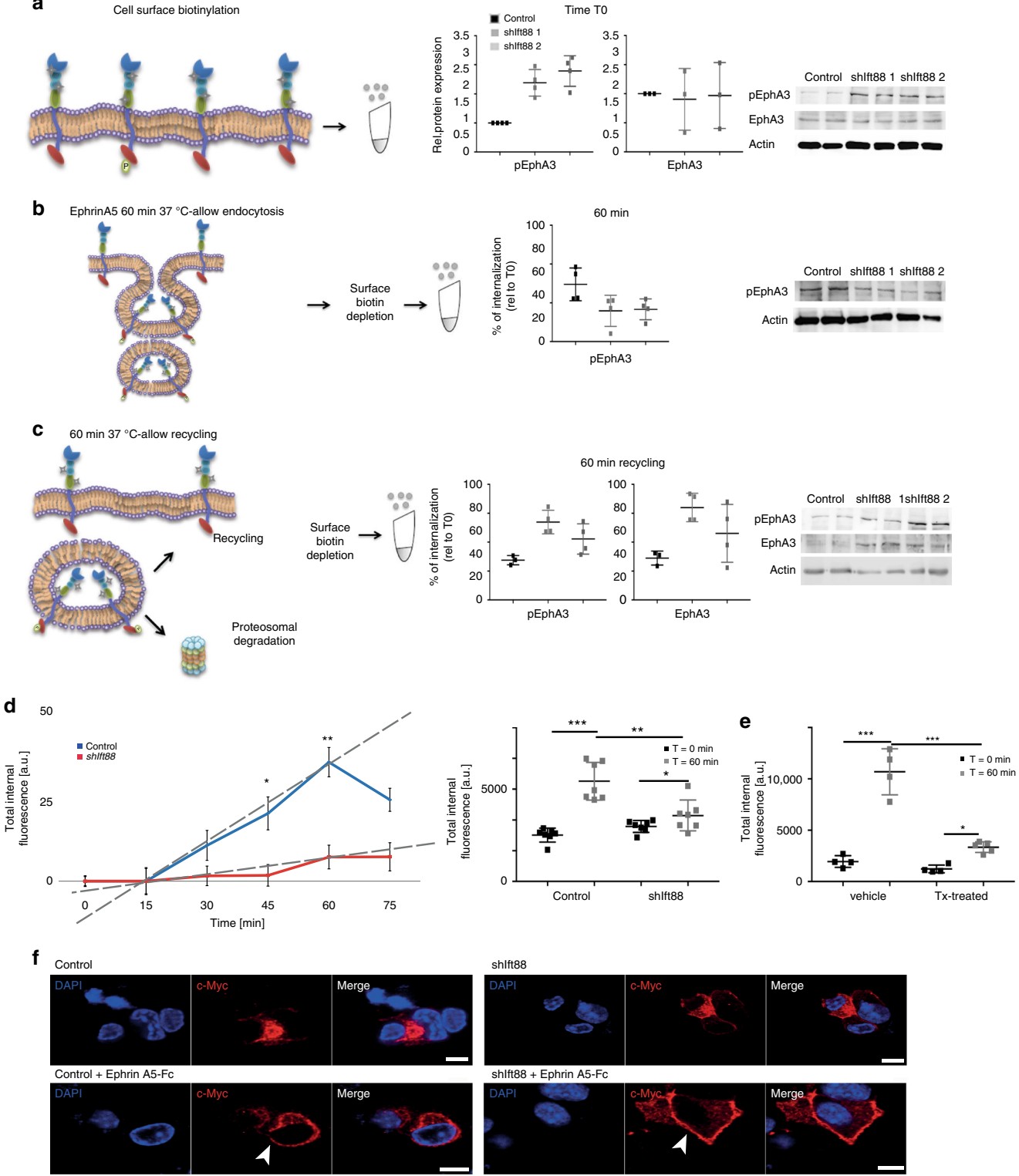

**Fig. 4 Vesicular trafficking to the endosomal recycling compartment depends on *Ift88* function. a** EphA3 and pEphA3 at the cell surface, mean ± s.d.
**b** Internalized pEphA3, mean ± s.d. **c** pEphA3 and EphA3 in the cytoplasm after internalization and before recycling back to the cell surface, mean ± s.d.
**d** EphrinA5 uptake in shIft88 cells. Biotinylated EphrinA5 was visualized with Streptavidin-AlexaFluor 647 ($t = 45$ min: $p = 0.0379$; $t = 60$ min: $p = 0.0062$;
$t = 75$ min: $p = 0.0329$ ($t$ test), mean ± s.d.), discontinuous lines show different slopes, on the side dot plots of $t = 60$ min compared with $t = 0$ min in both
cell lines. **e** EphrinA5 uptake in Tx- and veh-treated βICKO islets ($n = 3$, Δ(vehicle): $p = 0.0001$; Δ(treated): $p = 0.0132$; vehicle $t = 60$ min vs treated $t = 60$ min: $p = 0.0001$ ($t$ test), mean ± s.d.). **f** c-Myc-EphA3 (red) localization in *shIft88* expressing MIN6m9 cells with or without Ephrin A5-FC (1 h prior
experiment). Scale bar 5 μm.

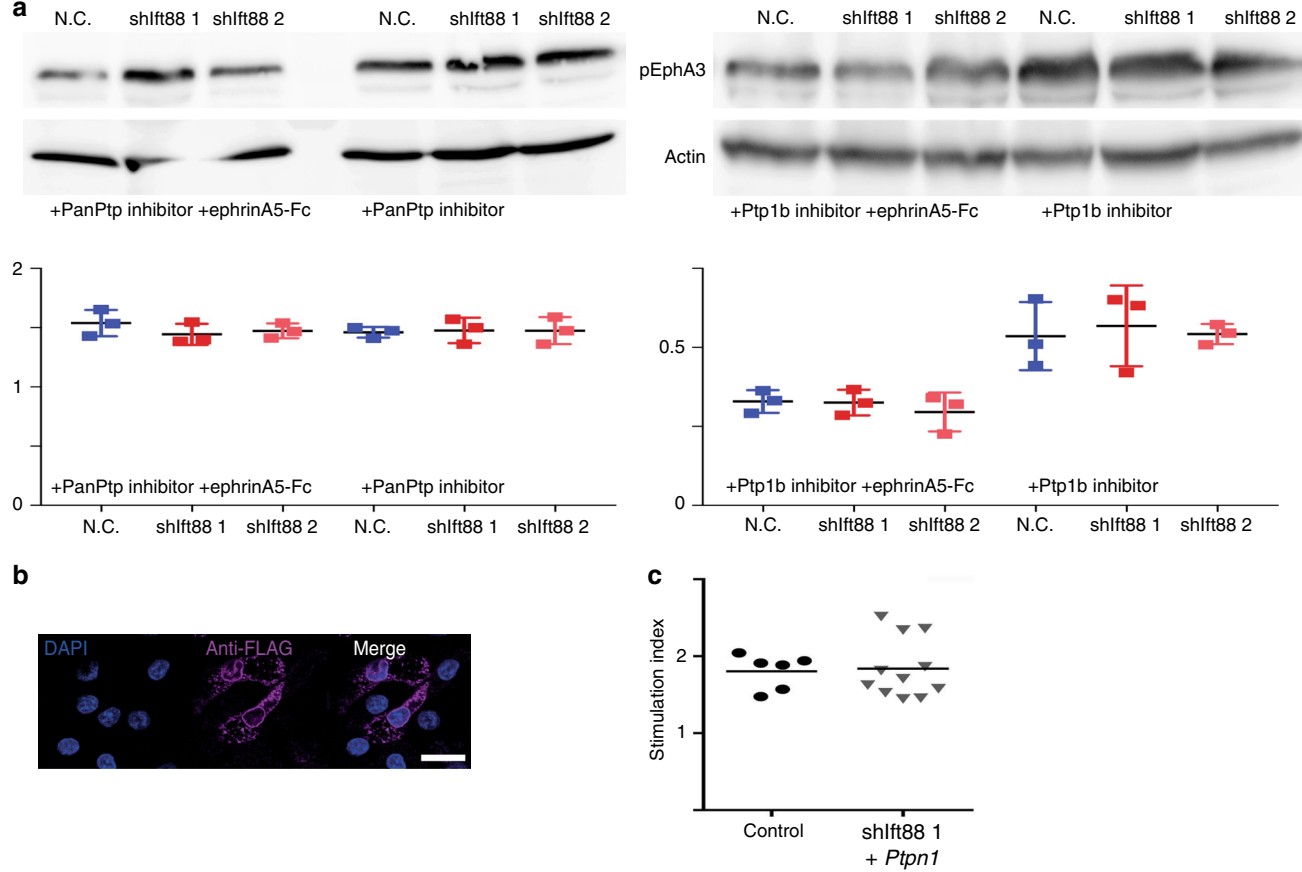

**Fig. 5 Ciliary dysfunction impairs phosphatase activity. a** pEphA3 (Tyr779) and actin levels in shIft88 1 and 2 and control cells treated with CinnGEL 2-methylester (Ptp1b-specific inhibitor) or PTP inhibitor II in absence or presence of 1 µg ml$^{-1}$ EphrinA5-Fc with quantifications, mean ± s.d. **b** Localization of Flag-tagged PTP1B (magenta) in MIN6m9 cells (scale bar 20 µm). **c** Insulin secretion of *shIft88* expressing MIN6m9 cells transfected with *Ptpn1* (2 µg plasmid per well) (*n* = 6 control, 12 shIft88 experiments, insulin measured *n* = 2 per experiment, mean ± s.d.).

only PTP that dephosphorylates EphA3. Notably, pEphA3 levels were independent of Ift88, suggesting differential recruitment of Ptp1b as the origin of the observed defect. When treated with EphrinA5-Fc and Ptp1b inhibitor, EphA3 phosphorylation is reduced. Ift88 had no additional effect under these conditions (Fig. 5a). EphrinA5-Fc treatment has no effect on pEphA3 levels when all PTPs are inhibited, suggesting that ligand binding recruits at least one additional PTP (Fig. 5a).

We transfected *shIft88* expressing MIN6m9 cells with FLAG-tagged *Ptpn1* cDNA. Overexpressed Ptp1b localized predominantly, but not exclusively, to the perinuclear region. We also observed Ptp1b in more peripherally located vesicular compartments (Fig. 5b). We challenged *Ptpn1* and *shIft88* expressing MIN6m9 cells with 16.7 mM glucose and found that they secreted insulin similar to control cells (Fig. 5c). This strongly implicates EphA hyperphosphorylation as the cause for decreased insulin secretion in β-cells depleted of Ift88. Taken together, these experiments suggest that ciliary impairment leads to a defect in EphA3 internalization, resulting in the suppression of spontaneous EphA3 phosphorylation.

***Ift88* is involved in maintaining epithelial-like polarity**. As stated above, EphA3 endocytosis is stalled before reaching the perinuclear compartment where Ptp1b is located. Immunofluorescent images of *EphA3-myc* expressing cells show that endocytic vesicles with EphrinA5-EphA3 complexes are enriched

in the cell periphery potentially indicating a problem with the actin network.

Because rearrangement of the actin cytoskeleton is one of the first steps in endocytosis, we first tested the ratio of polymerized actin (F-actin) to actin monomers (G-actin). In *Ift88*-depleted cells, we found increased levels of F-actin and an increased ratio of F- to G-actin (Fig. 6a). We used phalloidin staining to visualize F-actin (Fig. 6b) and found that *Ift88*-depleted cells show less long stress fibers and differently organized cortical actin. Actin fibers appear shorter and thicker in the absence of *Ift88*. To confirm a role for actin dynamics in EphrinA5-Fc-mediated EphA3 endocytosis, we treated cells with 2 µM cytochalasin-D to inhibit actin polymerization. Interfering with the dynamics of actin-based filaments ablates EphrinA5-Fc uptake irrespective of *Ift88* levels, demonstrating that actin reorganization plays an important role in receptor-mediated EphrinA5-Fc endocytosis (Fig. 6c). Actin reorganization is in part regulated by polarity. We and others have shown previously that primary cilia are involved in planar cell polarity (PCP) signaling[29,30]. Depletion of ciliary/basal body proteins is associated with increased canonical, β-catenin-dependent Wnt signaling[29,31]. In *Ift88*-depleted cells, β-catenin levels are increased by ~50% (Fig. 5d). Another important component of cellular polarity is T-cell lymphoma invasion and metastasis 1 (Tiam1), a GTP-exchange factor (GEF) for Rac1 that is regulating actin polymerization. Importantly, Tiam1 forms a complex with the Par3-Par6-aPKC complex regulating cellular

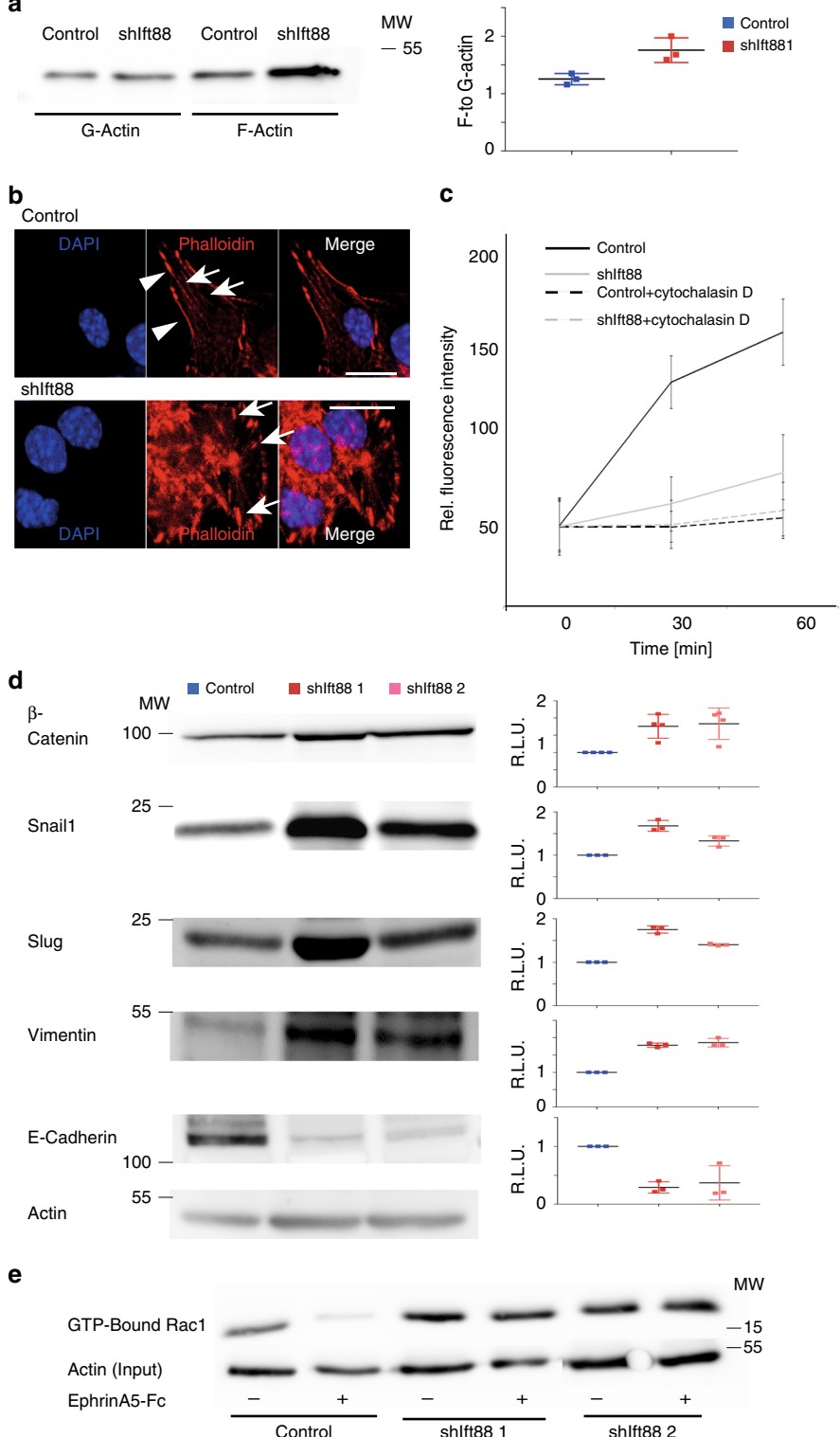

**Fig. 6 *Ift88* is involved in maintaining epithelial-like polarity. a** F- and G-actin quantification in in shIft88 1 and 2 and scrambled RNA (control), mean ± s.d. **b** Modification—actin visualized by phalloidin staining (red) in shIft88 cells and scrambled RNA (control) (scale bar 10 μm). **c** Biotinylated EphrinA5-Fc uptake in shIft88 cells treated with cytochalasin-D (2 μM, 3 h) and vehicle, mean ± s.d. **d** β-Catenin, Snail1, Slug, Vimentin, E-Cadherin protein levels (*n* = 3) in shIft88 1 and 2 cells and controls and quantification, mean ± s.d. **e** GTP-bound Rac1 levels in shIft88 1 and 2 and control cells treated with EphrinA5-Fc or vehicle.

polarity[32]. When testing Tiam1 mRNA and protein levels, we found that they were significantly increased in *Ift88*-depleted cells and Tx-treated βICKO islets (Fig. 7a), indicating a change in cell polarization. In addition, an increase of F- to G-actin is often observed in cells undergoing delamination[33]. Because β-catenin-dependent Wnt signaling can induce epithelial-to-mesenchymal-transitions (EMT)[34,35], we next checked for epithelial- or mesenchymal-like characteristics. Transcription factors Snail1 and Slug/Snail2 are induced by Wnt signaling and promote cell adhesion, cell migration, and metabolic reprogramming[36]. In *Ift88*-depleted cells, both Snail1 and Slug are increased (Fig. 6d). Similarly, vimentin, a filamentous protein characteristically upregulated in EMT and a downstream target of Slug is more than twofold increased (Fig. 6d). At the same time, E-cadherin, a target of Snail1 and Slug transcriptional repression is significantly downregulated (Fig. 6d). Taken together, loss of *Ift88* function leads to a transition toward mesenchymal-like cellular characteristics. This polarization defect could underlie differences in actin-polymerization observed in *Ift88*-depleted MIN6m9 cells.

**_Ift88_ is required for Ephrin-stimulated Rac1 activation**. Ligand internalization and vesicular trafficking are highly regulated[10]. Rac1 regulates a critical first step of endocytosis, re-organization of the actin cytoskeleton[37]. Moreover, Rac1 also localizes to early endosomes[10]. Because Tiam1 is the GEF for Rac1, we speculated Rac1 activity is affected. To test if misregulated GTP-loading is linked to the loss of EphrinA5-Fc stimulated Rac1 activity, we treated the cells with EphrinA5-Fc to trigger endocytosis and quantified the amount of GTP-bound Rac1 (Fig. 6e). Upon EphrinA5-Fc-stimulation, GTP-bound Rac1 levels decrease indicating GTP hydrolysis and thus Rac1-activity. By contrast, GTP-bound Rac1 is increased in unstimulated, *Ift88*-depleted cells. Stimulation with EphrinA5-Fc does not induce GTP hydrolysis, suggesting that EphrinA5-Fc related Rac1 activity is lost when ciliary function is compromised, and that reorganization of the actin cytoskeleton and early endocytosis are impaired (Fig. 6e).

In addition to protein levels, we also tested the localization of Tiam1 in *Ift88*-depleted cells. Tiam1 immunofluorescence is widely distributed throughout the cytoplasm and seemingly strongly increased (Fig. 7b). In control cells, Tiam1 is associated with the GM130-positive cis-Golgi network (CGN). To better quantify the change in cellular localization, we determined Pearson's correlation coefficient and observed a significant decrease in *Ift88*-depleted cells. This is indicative of a less strict spatial association between Tiam1 and the CGN compared with controls. This likely is newly synthesized protein coming from the ER waiting to be distributed to its subcellular destination. The cytoplasmic distribution of Tiam1 might simply be reflective of the relative overexpression shown in Fig. 7a, b.

To test if excess levels of Tiam1 are linked to decreased insulin secretion, we treated isolated βICKO islets with Tiam1 inhibitor NSC23766 ($N^6$-[2-[[4-(diethylamino)-1-methylbutyl]-6-methyl-4-pyrimidinyl]-2-methyl-4,6-quinolinediamine trihydrochloride)[38]. We expected to restore EphrinA5-Fc uptake and glucose-stimulated insulin secretion in Tx-treated βICKO islets to those of NSC23766-treated control islets, if Tiam1 was indeed involved. Insulin secretion from Tx-treated βICKO islets incubated with 20 μM NSC23766 was partially restored (Fig. 7c). Importantly, Tx-treated βICKO islets secrete similar levels of insulin as those of control islets, in which all Tiam1 activity is blocked. When pre-incubating *shIft88* expressing cells with 20 μM NSC23766, EphrinA5-Fc uptake was partially restored both with respect to rate and overall fluorescence intensity (Fig. 7d). Taken together,

our data suggest that increased Tiam1 activity is involved in the insulin secretion defect observed in Tx-treated βICKO mice (in vivo and ex vivo) and that this effect is mediated by partially blocking EphA/EphrinA5 internalization.

**_IFT88_ depletion disrupts insulin secretion in human islets**. Tx-treated βICKO mice have T2DM symptoms, including defects in insulin secretion and loss of β-cell mass. However, human and murine islet architecture and morphology are different[39,40]. Importantly, Ephrin ligand and receptors are expressed in human islets, which makes it more likely that our findings in mice have relevance to humans[41]. It is known that inhibition of Eph receptors enhances glucose-stimulated insulin release from human islets[42].

To probe relevance to human β-cells and potentially T2DM, we obtained human islets from the Alberta Diabetes Institute IsletCore[43]. Islets from four different normoglycemic donors were transduced with a lentiviral delivery vector expressing shRNA targeting *IFT88* or scrambled RNA and red fluorescent protein (RFP; Fig. 8a). Based on fluorescence, we estimate transduction efficiencies to be >50% and IFT88 protein levels were ~twofold reduced (Fig. 8b).

EPHA3 phosphorylation is doubled in *IFT88*-depleted islets of all islet preparations (Fig. 8b) In addition, *IFT88* depletion impaired glucose-stimulated insulin release in all four islet preparations (Fig. 8c)). Of note, the stimulation indices vary considerably, likely due to differences in age, metabolic state etc. of the organ donors. In all four samples, however, glucose-stimulated insulin release is significantly decreased. In good agreement with our data from Tx-treated βICKO mice, we did not observe differences when treated with membrane-depolarizing KCl. In summary, we suggest that IFT88 is important for EphA-processing and insulin secretion in both mouse and human islets.

To test if the observed effect could play a role in T2DM, we tested pEPHA3 levels in islets from three different diabetic donors (two females, one male donor, age 45–71 years; BMI = 22–38.3) and compared them to islets from normoglycemic donors (two females, age 48, 61 years; BMI 29.2/26, respectively). Phosphorylated EPHA3 was low in islets from normoglycemic donors (Fig. 8d, R268 and R271) as we had observed before (Fig. 8b). In islets from diabetic donors, however, we found pEPHA3 to be consistently upregulated (diabetic patient on average + 58%). This supports a role for EPHA-dependent signaling and possibly IFT88 and primary cilia function in glucose homeostasis and T2DM. To test for a more direct link between islet cilia/basal body function and Type 2 diabetes, we analyzed islet transcriptomes obtained from pancreatic resections of human donors. In a cohort (N = 19) of non-diabetic, prediabetic, and diabetic individuals[44], we found significant misregulation of *RAB3IP* (*RAB3A interacting protein*), a direct interaction partner of BBSome component BBS1 and a GEF of RAB8A[45]. *RAB3IP* mRNA levels associate negatively with increases in fasting glucose and positively with insulin secretion, i.e. HOMA2%B[46] calculated using C-peptide and insulin (Table 1; Supplementary Table S1). Although this finding tentatively supports a functional role for cilia/basal bodies in insulin secretion and islet function in humans, we have to emphasize the need for further validation of our findings. Due to a lack of patient RNA samples, we could not confirm the expression data by qPCR or another independent line of evidence. However, we suggest that our findings strongly support the case for a more detailed investigation of the role primary cilia and basal body genes and proteins might play in glucose homeostasis, insulin secretion, and T2DM.

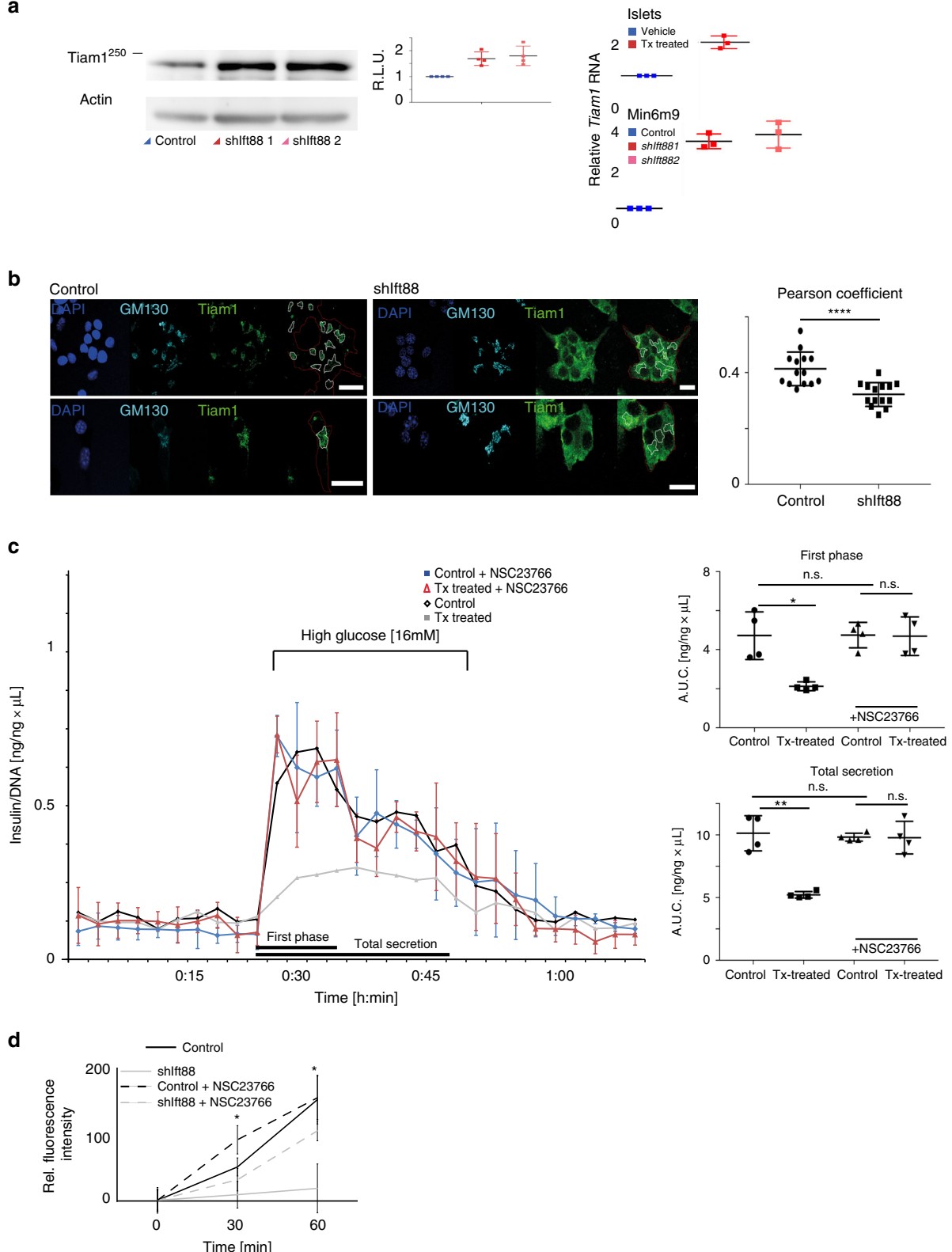

**Fig. 7 Tiam1/Rac1 misregulation inhibits insulin. a** Protein and mRNA levels of *Tiam1* in isolated βICKO islets treated with tamoxifen and vehicle and MIN6 cell lines, mean ± s.d. **b** Tiam1 (green) and GM130 (Cyan) localization in *shIft88* expressing MIN6m9 cells with Pearson coefficient ($p < 0.0001$, *t* test, mean ± s.d.). Scale bar 20 μm. **c** Dynamic insulin secretion of isolated βICKO islets treated with tamoxifen (blue) or vehicle (ethanol; red) treated with 20 μM NSC23766. On the right, quantification of first phase (AUC, time 18:00 to 30:00 min) and total secretion (AUC, from 18:00 to 48:00 min) (islets pooled from $n = 10$ animals, experiment repeated $n = 4$, and insulin measured $n = 2$ per experiment), in gray Tx-treated βICKO islets and in black vehicle-treated βICKO islets from Fig. 1f for comparison, mean ± s.d. **d** Biotinylated EphrinA5-Fc uptake in derivative MIN6m9 cells treated with NSC23766 and vehicle. ($n = 4$, $t = 30$ min: Control vs shIft88 $p = 0.0044$, Control vs shIft88 + NSC23766: $p = 0.0289$; $t = 60$ min: Control vs shIft88 $p = 0.0006$, Control vs shIft88 + NSC23766 $p = 0.1466$ (ANOVA), mean ± s.d.

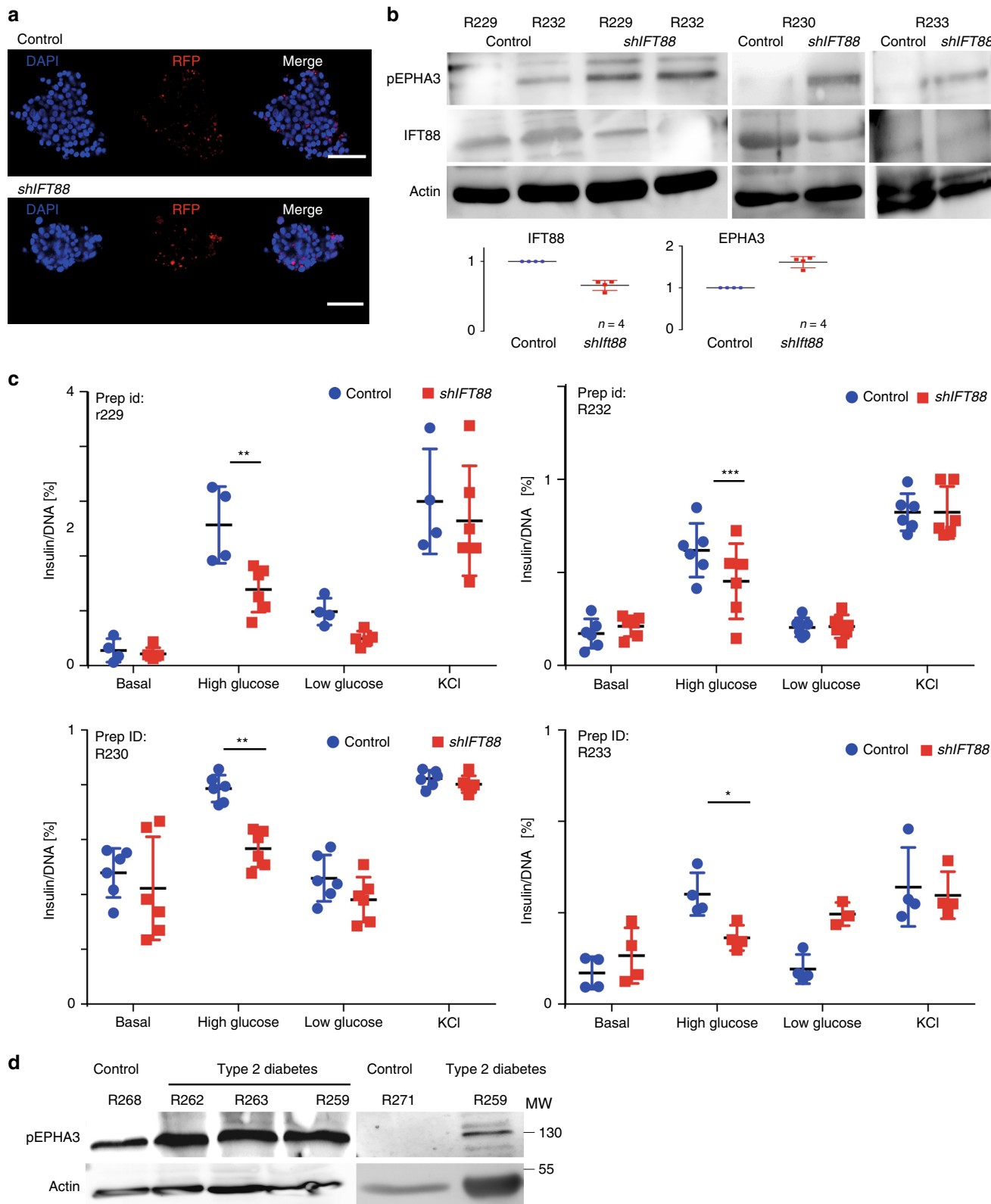

## Discussion

Here we show that primary cilia play a role in adult tissue homeostasis and β-cell maintenance and function. Ablating a core cilia component, *Ift88*, from β-cells in young adult mice leads to rapid deterioration of glucose tolerance, insulin secretion and, eventually, β-cell loss (Fig. 1; Supplementary Fig. 2). In βICKO

islets and insulinoma cells depleted of *Ift88*, we found that EphA receptors were hyperphosphorylated and that this elevation in pEphA levels blocked insulin secretion from induced βICKO islets (Figs. 2, 3). EphrinA5 does not undergo efficient endosomal transport in β-cells with impaired cilia function in part because of a misregulation of the Tiam1-Rac1 signaling axis that is involved

**Fig. 8 Ciliary deficiency and EphA hyperphosphorylation block insulin secretion in human islets. a** RFP (red) expression in human primary islets (scale bar 50 μm) **b** Immunoblotting of pEPHA3 (Tyr 779), IFT88, and actin protein levels in primary human islets, mean ± s.d. **c** Glucose-stimulated insulin secretion on human primary islets expressing *shIFT88* or scrambled RNA (N.C. RNA) from four different donors (prep ID:R229, $n = 6$, N.C.RNA high glucose: $2.5 ± 0.5$ ng ng$^{-1}$ μL$^{-1}$ insulin/DNA; *shIFT88* high glucose: $1.38 ± 0.37$ ng ng$^{-1}$ μL$^{-1}$ insulin/DNA; $p = 0.005$ (*t* test); prep ID: R232, $n = 6$, N.C.RNA high glucose: $0.64 ± 0.11$ ng ng$^{-1}$ μL$^{-1}$ insulin/DNA; *shIFT88* high glucose: $0.28 ± 0.11$ ng ng$^{-1}$ μL$^{-1}$ insulin/DNA; $p = 0.002$ (*t* test); prep ID:R230, $n = 6$, N.C.RNA high glucose: $0.79 ± 0.07$ ng ng$^{-1}$ μL$^{-1}$ insulin/DNA; *shIFT88* high glucose: $0.56 ± 0.12$ ng ng$^{-1}$ μL$^{-1}$ insulin/DNA; $p = 0.0023$ (*t* test); prep ID: R233, $n = 6$, N.C.RNA high glucose: $0.56 ± 0.13$ ng ng$^{-1}$ μL$^{-1}$ insulin/DNA; *shIFT88* high glucose: $0.36 ± 0.06$ ng ng$^{-1}$ μL$^{-1}$ insulin/DNA; $p = 0.02$ (*t* test) [**R229** = 22 y/o, F, BMI = 23.0; **R230** = 58 y/0, M, BMI = 29.4; **R232** = 66 y/o, F, BMI = 18.5; **R233** = 76 y/o, F, BMI = 19.3], mean ± s.d. **d** Immunoblotting of phosphorylated (Tyr 779), IFT88, and actin of primary human islets lysates [(**R268** = 29.2 y/o, F, BMI = 29.2; **R262** = 54 y/o, F, BMI = 22.; **R263** = 71 y/o, M, BMI= 38.3; **R259** = 45 y/o, F, BMI = 33.1; **R271** = 60 y/o, F, BMI = 26.0], mean ± s.d.

**Table 1 *RAB3IP* transcript is misregulated in islets from (pre-)diabetic donors.**

|  | Effect estimate | SD of effect estimate | *p*-value |
|---|---|---|---|
| Fasting glucose | −0.047 | 0.018 | 0.02 |
| HbA1c | −0.107 | 0.051 | 0.05 |
| HOMA2%B (C-peptide) | 0.005 | 0.002 | 0.03 |
| HOMA2%B (insulin) | 0.004 | 0.002 | 0.03 |

Inverse correlation of a *RAB3IP* transcript (238853_at) in human islets with fasting glucose, HbA1c, and positive correlation with insulin secretion (HOMA2%B, C-peptide, and insulin based) ($N = 19$)

in cell polarity, actin polymerization, and early endocytosis. Blocking Tiam1-Rac1 interaction restores EphrinA internalization and insulin secretion from βICKO islets.

Previous work from us and others suggested a role for primary cilia in β-cell function. In *Danio rerio*, suppression of bbs-proteins increases β-cell fragility[13] and we showed that *Bbs4*$^{−/−}$ mice have impaired glucose handling and insulin secretion before onset of obesity[4]. To better understand the role of primary cilia function in β-cells, we ablated cilia specifically from adult β-cells by Tx-treatment of βICKO mice. As early as 4 weeks post induction, we observed significantly impaired glucose handling that deteriorated further over time (8 and 12 weeks post induction; Fig. 1; Supplementary Fig. 2). Similar to *Bbs4*$^{−/−}$ mice, acute insulin release was attenuated in βICKO mice after i.p. injection of glucose 8 weeks post induction and deteriorated further. At 6 weeks post induction, the β-cell mass of Tx-treated βICKO mice is similar to that of vehicle-treated controls; therefore, loss of β-cell mass does not explain the defect in insulin secretion. At the end of our long-term experiment, we found a roughly sixfold reduction in β-cell mass of Tx-treated βICKO mice vs vehicle-treated controls, possibly due to persistent ERK/MAPK activation that has been shown detrimental for β-cell maintenance and survival[47,48]. We previously published that acute depletion of Bbs4 protein from murine islets impairs first-phase insulin secretion[4]. Importantly, another ciliopathy model, *Bardet-Biedl Syndrome 5* (*Bbs5*$^{−/−}$) knockout mice have significantly impaired glucose tolerance at 13 weeks of age (ref.[49]; www.mousephenotype.org). βICKO mice manifest impaired insulin secretion in combination with a loss of β-cell mass similar to that reported for Type 2 *Diabetes mellitus* patients[50].

In *Bbs4*$^{−/−}$ and Tx-treated βICKO islets, EphA2 and EphA3 phosphorylation is upregulated by >100% and downstream signaling node ERK/MAPK is roughly twofold overactivated (Fig. 2). Persistent activation of the ERK/MAPK signaling pathway has been shown to induce apoptosis and β-cell death in pancreatic islets[47,48], which could explain the loss of β-cell mass in our model 20 weeks post induction.

Several lines of evidence support a role for Ift88 and cilia function in EphA-processing: (A) treatment with EphrinA5-Fc reduces EphA phosphorylation levels and restores insulin secretion to the levels of control islets or insulinoma cells; (B) over-expression of a dominant negative isoform of EphA5 (DN-EphA5) partially rescues glucose-stimulated insulin secretion in *Ift88*-depleted MIN6m9 cells; (C) overexpression of human *Ptpn1*, encoding for Ptp1b, one of the main negative regulators of EphA receptors, also restores insulin secretion in *Ift88*-depleted MIN6m9 cells.

We show that the amount of EphA3 presented at the cell surface is similar in presence or absence of Ift88. Importantly however, pEphA3 is increased at the plasma membrane in absence of Ift88 and internalization upon EphrinA5-Fc stimulation is delayed (Fig. 4). Upon ligand binding, EphA/EphrinA complexes are sorted into early endosomes and targeted for endosomal recycling. Importantly, we also observed a defect in receptor recycling back to the cell surface; a concomitant increase in pEphA3 levels indicates that one or more early steps in endosomal processing—prior to the perinuclear endosomal recycling compartment—are perturbed in ciliary dysfunction. EphrinA5-Fc internalization is slower and does not reach as many endosomal compartments when Ift88 is ablated (Fig. 4). Moreover, our data implicate recruitment of at least one additional protein tyrosine phosphatase to ligand-bound EphA receptor, suggesting that endosomal processing of occupied EphA receptor is different from that of ligand-free EphA (Fig. 4; Supplementary Fig. 5).

One of the earliest steps of endocytosis requires a rearrangement of the actin cytoskeleton; we found that actin polymerization is altered in *Ift88*-depleted cells and that this might be linked to a change in cellular polarization. We observed increased levels of EMT markers Snail, Slug, and Vimentin in *Ift88*-depleted MIN6m9 cells and that E-cadherin is decreased, consistent with EMT processes. Rac1 is a major regulator of polarity-dependent actin reorganization, and we found that GTP-bound Rac1 is increased in Ift88-depleted MIN6m9 cells; moreover, addition of EphrinA5-Fc does not induce GTP hydrolysis as was observed in control cells. Tiam1, a polarity protein and one of the Rac1-GEFs, is elevated in Ift88-depleted cells and Tx-treated βICKO islets (Fig. 5). Treatment with NSC23766, a Tiam1/Rac1 inhibitor, restores insulin secretion from Tx-treated βICKO islets to the level of controls treated with NSC23766. Tiam1 inhibition also partially recovers EphrinA5-Fc uptake in *shIft88*-expressing MIN6m9 cells. Possibly, Tiam1 occupies binding sites that are rendered unavailable to other Rac1 effectors required for signal propagation. Previous studies have found excessive GTP-bound RhoA levels in primary kidney cells of *Bbs4*-, *Bbs6*-, and *Bbs8*-null mice; treatment with ROCK inhibitors restores ciliation to levels observed in wt control cells[51]. We suggest that ciliary function is required for polarity signaling and reorganization of the actin skeleton and that blocking rearrangement results in failure to negatively regulate EphA-autoactivation via endosomal transport.

Misregulation of EphA-activity in β-cells results in loss of insulin secretion and severely impaired glucose homeostasis in Tx-treated βICKO mice.

Finally, we have shown evidence implicating cilia/basal body function in insulin secretion in humans:

When we depleted islets from four different donors of *IFT88*, we observed increased EphA3 phosphorylation levels and impaired glucose-stimulated insulin secretion, both in good agreement with our findings in Tx-treated βICKO mice. We observed increased pEPHA3 levels in islets from diabetic donors compared with normoglycemic controls. Misregulation of *RAB3IP* in islets from diabetic donors tentatively supports a direct link between cilia/basal body dysfunction, vesicle trafficking, and Type 2 *Diabetes mellitus*; still, further studies are warranted to more firmly establish such a link. Taken together, we propose that modulation of ciliary function and endosomal signal processing could offer a novel approach to therapeutic interventions in metabolic diseases such as Type 2 *Diabetes mellitus*.

## Methods

**Ethical approval**. Experimental procedures involving live animals were carried out in accordance with animal welfare regulations and with approval of the Regierung Oberbayern (az 55.2-1-54-2532-201-15 islets) and (az 55.2-1-54-2532-187-15 Tamoxifen). Experiments involving lentivirus and biosafety level 2 requirements were registered with the Regierung Oberbayern (az 50-8791-8.1005.2021). Human primary islets from normoglycemic and diabetic donors were supplied by the Alberta Diabetes Institute IsletCore (Edmonton, Canada), supported by the Alberta Diabetes Foundation (ADF), the Human Organ Procurement and Exchange (HOPE), and the Trillium Gift of Life Network (TGLN) for coordinating donor organs the Islet Core. Because the donors were deceased at the point of organ collection, the family gave informed consent for use of pancreatic tissue in research. The Human Research Ethics Board at the University of Alberta approved the study (Pro00001754, Pro00013094, ref. [43]). Ethical approval for the use of pancreatic tissue from the IsletCore was obtained from the Human Ethics Research Board in Munich (Az 557/16S). The collection of human material from PPP and the study were approved by the Ethics Commission of the Medical Faculty of the University of Tübingen (#697/2011BO1 and #355/2012BO2).

**Induction of recombination in βICKO Mice**. Between P25 (post-natal day 25 and P35), *Pdx1-Cre^ER; Ift88^{loxP/loxP}* (βICKO) with mixed genetic background of C57BL/6j and C3H/HeJ were induced by oral gavage (100 mg/kg body weight of tamoxifen dissolved in corn oil) once per day on 5 consecutive days.

**Islet isolation**. To isolate the islets of Langerhans from the pancreas, we injected in the bile duct a solution of 1 mg/ml of Collagenase P (Roche) in G-solution. In total, between 2 and 3 ml of solution were injected. After that the pancreata were carefully removed to avoid damages and placed in glass vial containing other 3 to 4 ml of Collagenase P solution. The pancreata were then incubated in water bath at 37 °C (Lauda) for 15 min. After 7.5 min, the vials were vigorously shaked to break the pancreata into pieces. To block the digestion process, the vials were moved in ice and 10 ml of G-solution were added. To clean the islets of Langerhans from the endocrine tissue, two steps were performed: first we took advantage of the density and weight of the islets that sediment to the bottom of the vial. The supernatant was collected and discarded, and other 10 ml of G-solution were added every 5 min. The process was repeated four to five times or until the supernatant was clear. The second step of the cleaning consisted in sequential hand-picking of the islets of Langerhans with a 200-µl pipette (Eppendorf) in a dishes with clean G-Solution. The washing was performed three to four times, depending on the state of the culture. As a last step, the islets were hand-picked and placed in culture medium.

**Ex vivo induction of CreER**. Isolated islets were plated in 10 cm low attachment dishes in culture medium. Twenty-four hours after, they were treated with 1 µM Tamoxifen in ethanol. Medium was changed back to normal culture medium 24 h after treatment. After 5 days, they were used for experiments.

**Glucose and serum insulin level in mice**. Animals were fasted for 12 h before tests. We injected 2 g/kg of a 20% glucose solution intraperitoneally. Blood glucose was measured with Contour xt (Bayer). Blood serum was collected, and serum insulin was measured with a mouse plasma insulin HTRF kit (62IN3PEF, Cisbio).

**Insulin measurements**. Quantifications of insulin content were carried on stimulated cells or islets using Insulin HTRF kit (62INSPEC, Cisbio) according to the manufacturer's instruction, for human islets has been used Insulin ELISA (10-1113-01, Mercodia) according to the manufacturer's instruction.

**Cell and tissue culture**. MIN6m9 cells were cultured in DMEM (D6046, Sigma) supplemented with 10% fetal bovine serum (12103C, Sigma), 1% penicillin/streptomycin (15140-122, Gibco), 20 mM D-Glucose (HN06.4, Roth), and 10 µl/L β-Mercaptoethanol (31350-010, Gibco). Isolated islets were cultured in RPMI-1640 (21875-034, Gibco) supplemented with 10% FBS and penicillin/streptomycin.

**Immunofluorescent stainings**. Cells or islets were fixed in 4% paraformaldehyde (37 °C, 20 min). Permeabilized in 0.1% (cells or islets) or 0.2% (pancreas section, for fixation see above) Triton-X in PBS (30 min) and block in 5% FBS (cells) or 10% FBS, 3% donkey serum (Islets and Pancreas sections) and incubate overnight at 4 °C with primary antibody diluted in blocking solution.

Primary antibodies used: Nkx6.1 (goat, R&D System, AF5857, 1:300), Caspase-3 (rabbit, Cell Signaling, 9664, 1:300), Ki67 (rabbit, Abcam, ab15580, 1:300), acetylated α-tubulin (mouse IgG2b, Sigma, T6793, 1:2500 cells, 1:1000 islets), Flag (rabbit, Sigma, F7425, 1:500), GM130 (mouse, BD biosc., 610822, 1:300), EphA5 (rabbit, Santa Cruz, sc-927, 1:500), Tiam1 (rabbit, Santa Cruz, sc-872, 1:500), c-Myc (Sigma, mouse, M4438). Cells, islets, or pancreas sections were then washed with PBS and incubated for 1 h (cells and islets) or 2 h (pancreas sections) with secondary antibody diluted in blocking solution at RT. Secondary antibody: anti-rabbit (488 nm, Invitrogen, A21206, 1:300 pancreas section, 1:2000 cells), anti-goat (633, Invitrogen, A21082, 1:300), anti-mouse IgG2b (647, Life Technologies, A-21242, 1:2000), anti-mouse (Cy5, Dianova, 715-175-151, 1:2000). Later, cells were incubated 10 min with DAPI (1:10000) and Phalloidin546 (Invitrogen, A22283, 1:300), Islets and Pancreas section only with DAPI. Cells, islets, and section were then washed three times in PBS and then embedded with Elvanol.

For brain immunofluorescent staining in the brain of *Pdx1-CreER;ROSA^{mT/mG}* induced with tamoxifen were dissected, fixated in PFA 4% for 48 h, passed to sucrose 10%, 20 and 30% solutions ON each, the brains were embedded in OCT (Leica, 14020108926) and frozen. In all, 20 µm cryosections were fixated with PFA 4% 30 min at RT, permeabilized with PBS-Triton 0.2% for 30 min and blocked for 1 h with 3% donkey serum. The cryosections were incubated ON with a dilution anti-chicken GFP 1/1000 (Aves Lab GFP-1020), rat anti-RFP 1/1000 (Cromoteck ORD003515), glucagon (Santa Cruz, Rabbit,sc-13091), insulin (Cell Signaling, Rabbit, 3014) in 3% donkey serum and 1 h at RT with a donkey anti-chicken-Cy2 (Dianova 703-225-155) 1/800 and donkey anti-rat-Cy3 (Dianova 712-165-153) in 3% donkey. The nucleus was identified as DAPI staining. Picture were taken with confocal microscope Leica Sp5 at magnification of 63 × 1.3 Glycerol (cells) or 20 × 0.75 immersol (islets) or with Nanozoomer 2.0 HT(Meyer) for pancreas section.

**EphrinA5-Fc uptake assay**. In total, $1 \times 10^5$ cells were seeded the day before the experiment in 24-well plate with coverslips. The day of experiment cells were treated with Recombinant EphrinA5/FC Chimera Biotinylated Protein (R&D System, BT374, 1:500) for different times. Cells were then washed and fixed as state above. EphrinA5/FC Chimera Biotinylated Protein was detected with Streptavidin, AlexaFluor 647 conjugate (Life Technologies, S21374, 1:1000) used as a secondary antibody.

**Image analysis**. Counting of proliferation, apoptosis, beta cell mass, and cilia number was done using IMARIS software (Bitplane). Analysis of fluorescent intensity in MIN6m9 for Transferrin-AF647 or Streptavidin-AF647 was done using IMAGEJ.

**Perfusion assay on islets**. To perform dynamic islet perfusion, we used a perifusion system (Biorep, PERI4-02) and we used 50 islets per column stratified by size (per column 10 small (<100 µm), 10 big (>200 µm), and 30 medium (between 100 and 200 µm) islets) immobilized according to the manufacturer's instruction (flow rate 100 µl/min, step length 2 min). Insulin secretion was normalized to DNA from the isolated islets. Islets were treated with EphrinA5-Fc (R&D System, 7396EA) 2 h before experiment or overnight (NSC23766 (Sigma, SML0952)).

**Western blots**. Cells or islets were lysed in RIPA buffer supplemented with protease (Thermo Scientific, 88265) and phosphatase (Thermo Scientific, 88667) inhibitor. Protein content was measured according to Lowry. Protein extracts were separated on a 10% polyacrylamide gel and transferred to nitrocellulose membranes with semi-dry transfer. Blots were probed with γ-tubulin (T6557, Sigma, 1:2500), Ift88 (13967-1-AP, Proteintech, 1:2000), Phospho EphA3 (8862, Cell Signaling, 1:1000), Pan-EphA3 (7038, Abcam, 1:1000), Ephrin B (3481, Cell Signaling, 1:1000), Phospho Erk (4377S, Cell Signaling, 1:1000), Pan Erk (4695, Cell Signaling, 1:1000), Phopsho Akt (4058S, Cell Signaling 1:1000), Pan Akt (2920, Cell Signaling, 1:1000), Phospho PI3K (4228, Cell Signaling, 1:1000), Pan PI3K (13666, Cell Signaling, 1:1000), Actin (612656, BD, 1:2500), Tiam1 (sc-872, Santa Cruz, 1:1000), Beta catenin (610154, BD,1:2000), Snail (C15D3, Cell signaling, 1:1000), Slug (PRS3959, Sigma, 1:1000), Vimentin (MAB3400, Merck, 1:2000), E-Cadherin (3195, Cell Signaling, 1:1000). Secondary antibodies: anti-mouse HRP (115-036-062, Dianova, 1:10,000), Anti-Rabbit HRP (111-036-045, Dianova, 1:10,000), the blots were then scanned with ChemStudio SA² (Analytik Jena) and analyzed on IMAGEJ; for phosphorylated vs pan protein were used: 680LT Anti Mouse (926-68022, Li-Cor, 1:10,000), 800CW anti-Goat (926-32214, Li-Cor, 1:10,000), 800CW

anti-Rabbit (926-32214, Li-Cor, 1:10,000) and scanned with Odissey Sa (Li-Cor) and analyzed with Image Studio. Cells were treated with ephrinA5-Fc (R&D System, 7396EA) overnight, PTP inhibitor II (Santa Cruz, SC202782) overnight, CinnGEL 2-methylesther (Santa Cruz, SC205633) overnight, or a combination of the compounds.

**Human islets culture**. Upon receipt human primary islets were culture in CMRL-1060 (11530037, Gibco) + with 10% human serum (4522, Sigma), 2 mM L-glutamin (25030081, Gibco) and 1% PenStrep (15140-122, Gibco) in an ultralow attachment plate (3262, Corning).

**Differentially expressed genes in human islets**. Islet tissues of human pancreas from pancreatic resections were excised using laser capture microscopy (PALM MicroBeam, Zeiss), as described previously (Gerst et al. JCEM, 2018). Briefly, the total RNA was extracted from collected islet tissue and subjected to transcriptome analysis using Human Genome Array HG_U133Plus2.0. Glucose, insulin, C-peptide, and HbA1c levels were measured from fasting blood samples collected before surgery. Insulin-secretion indices were computed as HOMA2%B both from insulin and C-peptide (Levy et al. Diabetes Care, 1998). Association of mRNA expression with glycemic traits was tested with linear regression models. The study was approved by the Institutional Review Board of the Medical Faculty of University of Tübingen (reference #697/2011BO1 and #355/2012BO2). Written informed consent was obtained from all participants.

**Reporting summary**. Further information on research design is available in the Nature Research Reporting Summary linked to this article.

## Data availability

Raw data and the significant blots can be found at: https://doi.org/10.6084/m9.figshare.7994213.v1.

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

## Acknowledgements

We would like to thank all members of the Gerdes lab and the Institute for Diabetes and Regeneration Research for their helpful suggestions, technical expertise, and discussions. We thank Noah Moruzzi for the generation of *shIft88* expressing plasmids and lentivirus. The *Ptpn1* cDNA plasmid was a generous gift from S. Paoluzi, University of Tor Vergata, Rome, Italy, and the *DN-EphA5* was a gift from E. Lammert, Heinrich-Heine Universität, Düsseldorf, Germany. Part of this work was funded by the German Center for Diabetes Research (J.M.G. and A.Z.). Thank you to Ingo Burtscher and Per-Olof Berggren for critical evaluation of the paper. We thank Aimee Bastidas-Ponce, Marta Tarquis, and Mostafa Bakhti for sharing *Pdx1-CreER;ROSA26*$^{mTmG}$ mice with us. We would like to thank Dr. Silvia Wagner (Department of General, Visceral and Transplant Surgery, University Hospital Tuebingen), Dr. Louise Fritsche, Andreas Vosseler, and Anja Dessecker (HMGU/IDM Tuebingen) for patient recruitment and study management. We thank Dr. Patrick MacDonald, Dr. Jocelyn Manning Fox, and James Lyon at the Alberta Diabetes Institute IsletCore (Edmonton, Canada) for the provision of human islets for research, supported by the Alberta Diabetes Foundation (ADF); the Human Organ Procurement and Exchange (HOPE) and the Trillium Gift of Life Network (TGLN) for coordinating donor organs. We wish to express our sincerest gratitude to all organ donors and their families for their generous gift to medical research. This study was supported by a grant (01GI0925) from the German Federal Ministry of Education and Research (BMBF) to the German Center for Diabetes Research (DZD e.V.).

## Author contributions

F.V. and A.S. characterized the phenotype of βICKO mice, isolated islets, and cultured and induced them in vivo and ex vivo. F.V. and N.O. ran immunoblots, F.V., N.O., and A.S. did immunostaining, fluorescence imaging and image analysis of βICKO pancreata and MIN6m9 cell lines. M.J.S. optimized and ran the pulse/chase surface biotinylation assay. F.V. and J.M.G. manipulated ex vivo insulin secretion. A.F. and R.W. recruited patients and obtained tissue and blood samples. F.G., S.U., and R.W. performed the transcriptome experiments and analyzed data. H.-U.H., S.U., and R.W. conceived and designed the transcriptome study. A.Z. advised the experimental design. All authors contributed to the paper. F.V. and J.M.G. wrote the paper. J.M.G. had the idea and designed the experiments.

## Competing interests

The authors declare no competing interests.
