## [Peer Review File · Nature Communications]

Reviewer #1 (Remarks to the Author):

Volta et al report a notable study on the role of cilia in EphA receptor tyrosine phosphorylation as well as insulin secretion, islet cell survival and human type 2 diabetes. Only one paper has been published to date on the role of EphA receptors and cilia, and no publication exists on the link between cilia, EphA receptors and islet function. The authors first provide evidence that mouse pancreatic islets with no cilia have increased activation of EphA2 and EphA3, and this could be demonstrated in two different cilia-defective mouse models. They then showed that manipulating the EphA receptor activation pathway rescued the defects observed in cilia-defective mouse islets and cilia-defective mouse insulinoma cells. They provide evidence showing that the cilia-defective islet cells have a defective endocytosis of EphA receptors, thereby preventing their dephosphorylation via PTP1b. The endocytosis defect is shown to be mediated via cilia-governed Tiam1/Rac1 signaling. Finally, the authors show that inhibition of cilia in human islets also results in EphA receptor hyperphosphorylation and that islets from subjects with type 2 diabetes have higher EphA3 tyrosine phosphorylation levels. The statistical analyses seem fine.

In sum, this is a very comprehensive and innovative study making its points using multiple different experimental systems ranging from cell line to mouse to humans. The observation of increased EphA3 activation in islets from humans with type 2 diabetes makes this story not only interesting from a cell biological point of view, but also for scientists working on clinical aspects of this widespread human disease.

A few minor issues still need to be addressed:

1. Two manuscripts have been published in "Diabetes", showing that EphA-ephrinA signaling affects glucagon release. It would be nice to see whether cilia also affect glucagon release, since Pdx1-CreERT mediated cilia defects shall also be present in alpha cells, not just in beta cells.
2. The histograms shall be shown as dot plots, which enable the reader to identify possible subgroups that might otherwise be hidden.
3. Several Western blots are overexposed. Even though this looks nice, a more scientific way is to show non-saturated Western blots wherever possible.
4. The authors shall describe in the M&M section how they obtained islets from humans with type 2 diabetes, as this is currently missing.

Reviewer #2 (Remarks to the Author):

The manuscript by Volta et al. addresses the role of primary cilia in adult pancreatic β -cell function. The authors deleted a core cilia component, Ift88, from β -cells in adult mice using an inducible Pdx1-Cre line and, subsequently, characterized glucose metabolism and β -cell function. The KO animals showed impaired glucose tolerance, defects in insulin secretion and, eventually, β -cell loss. Moreover, they went further to dissect the cellular mechanism(s) downstream of Ift88 in the β -cell, which would be responsible for the mediating insulin secretion. The authors found that EphA receptors are hyperphosphorylated in both KO islets and insulinoma cells depleted of Ift88 and showed that blocking the EphA signaling rescues the Ift88-knockdown insulin secretion phenotype. Finally, they provide some evidence that EphrinA does not undergo efficient endosomal transport in β -cells with impaired cilia function because of misregulation of the Tiam1-Rac1 signaling axis. Interestingly, IFT88 function and EPHA regulation seem to be conserved in human islets of control and diabetic patients.

Overall, this is a very interesting study reporting a novel signaling axis downstream of primary cilia in pancreatic β -cells. The biochemical evidence between Ift88 and EphA receptors is convincing, but the downstream connection with the Rac1-actin cytoskeleton is a bit tenuous and sometimes confusing – this could be expanded or removed. Additional specific comments are listed below.

Specific comments:

- 1) The data obtained in vivo from the β ICKO animals and ex vivo KO islets are quite convincing, including GTT, insulin secretion and perfusion assays. Less convincing are the results obtained in the rescue experiments with recombinant EphrinA5-Fc chimera (Fig. 3) as well as with Tiam inhibitor NSC23766 (Fig. 5). The perfusion experiment shown in Fig. 1 shows a dramatic difference in insulin secretion between control and the β ICKO islets, which cannot be appreciated anymore in the rescue experiments. Why the authors have not always included the untreated

control islets as baseline? This might help to appreciate the differences between control and mutant versus the rescued insulin secretion.

2) The differences in phosphorylation shown by WBs with the p-EphA3 antibody in sh- Ift88 knockdown MIN6 line treated with EphrinA5-Fc or Ptp1b inhibitor are not convincing, given the images shown in Fig. 3D and Fig.4 G. Are these images/ gels overexposed? This is an important point in particular with regard to the EphrinA rescue.

3) Does the increased phosphorylation affect only EphA2 and EphA3 receptors or all EphA receptors? Why the authors characterize only EphA3 and nothing is shown about EphA2?

4) The IF analysis is important and should be expanded and improved. First, why in the IF staining the authors used an antibody against EphA5 instead of EphA2 or EphA 3 (Fig. 3F) ? See also point above, is the effect specific ? Second, what is shown in the IF images? Single cells or clusters? What do the dotted lines demarcate? Third, EphA5 staining seems increased in the sh-Ift88, but the authors stated in the Result section that the effect is only on phosphorylation and not on total protein level. How do they explain this discrepancy? The IF should be supported by quantitative analysis and co-localization between GM130 and EphA should be measured.

5) The connection of Ift88 and Rac-actin cytoskeleton is not convincing. For ex. no data is included about the actin cytoskeleton organization or manipulation of actin polymerization.

6) The writing of the manuscript could be improved. Both Introduction and discussion are long and wordy, the important messages do not come through and it is sometime difficult to appreciate the novelties compared to previous literature on cilia in beta-cells.

Reviewer #3 (Remarks to the Author):

The paper demonstrated that adult-onset ciliary dysgenesis in the pancreatic beta-cells induced defective insulin secretion in response to glucose, especially the first-phase GSIS, and later, loss of beta cells. As for the molecular mechanisms underlying this phenomenon, they suggested that EphA hyperphosphorylation at the plasma membrane, due to defective endocytosis and phosphatase-mediated degradation of EphA-ephrin A5 complexes. They also suggested the increased Tiam1-Rac1 activity as a molecular mechanism for reduced endocytosis in cilia-defective beta-cells. These findings are novel and interesting in that they have firstly demonstrated a functional link between primary cilia and Eph-ephrin signaling and abnormal Eph signaling in human pancreas from diabetic subjects. However, they were considerably limited by a lack of control and statistics, small n numbers, and lots of errors in the manuscript and figures.

Major comments

1. To exclude any possible effects of tamoxifen on beta-cells and insulin secretion, another control group (Pdx-Cre negative, and tamoxifen-injected) was needed in Figure 1.
2. The n numbers of dynamic insulin secretion experiments in Fig 1F, 3A, and 5D were only 2. How many times were these experiments repeated to conclude the findings?
3. How does ciliary dysgenesis lead to increased Tiam-1 expression? The molecular link mediating this changes needs to be suggested or discussed.
4. The authors showed increased EphA phosphorylation in islets from the subjects with diabetes. However, they did not show ciliary changes in these subjects. Is EphA hyperphosphorylation linked to ciliary changes in human islets?

Minor comments

1. In Fig 2a, total EphA blot was missing. Fig 2A legend indicated phosphorylated p44/42 MAPK but p44/42 MAPK blot were also missing.
2. Fig 2B-D demonstrated the successful knockdown of IFT88 with shRNA transfection. These data should be presented as a Supplementary data.
3. In Fig 2F the first blot, the protein loading control i.e. pan Erk blot is needed.
4. Fig 3A and 5B had three groups. So AUC graphs should have three groups. Both tamoxifen-

untreated and ephrinA5-Fc or NSC23766-untreated control is needed.

5. Fig. 3C, x axis should have 4 groups. Moreover, data of low and high glucose condition should be shown in rectangles but there were only circles and triangles in the figure.

6. Incomplete sentence in page 9 line 123-125.

7. Page 10, line 142, Primary cilia are not just involved in insulin signaling. Is this description true? The previous paper (ref 5) has shown impaired insulin signaling by ciliary disruption.

8. In Fig 3C, 3D, and 3E, the description in the text and figures are mismatched.

9. The discription about Fig. 3B was ahead that of Fig.3A in the text.

10. Fig. 4F is missing in the text.

11. Control without treatment with PTP inhibitor is needed.

12. In Fig. 5 A-C, there is no statistics in the figures. Some groups had no error bars.

13. Fig 1A, repeated ANOVA rather than t-test should be performed to compare two groups each time point.

14. In Fig. 3F, 4F and 5F, quantification is needed.

Point-by-point response to our reviewers:

We thank the reviewers for the time and effort invested in our work, we truly feel that their comments helped improve the story and make this a better manuscript. We hope they will agree and find the manuscript revised to their satisfaction.

Reviewer 1:

In sum, this is a very comprehensive and innovative study making its points using multiple different experimental systems ranging from cell line to mouse to humans. The observation of increased EphA3 activation in islets from humans with type 2 diabetes makes this story not only interesting from a cell biological point of view, but also for scientists working on clinical aspects of this widespread human disease

A few minor issues still need to be addressed:

1. Two manuscripts have been published in "Diabetes", showing that EphA-ephrinA signaling affects glucagon release. It would be nice to see whether cilia also affect glucagon release, since Pdx1-CreERT mediated cilia defects shall also be present in alpha cells, not just in beta cells.

Thank you for this valuable comment. Indeed, EphA/ Ephrin A signalling has been reported to regulate glucagon release. However, at the time of induction, the *Pdx1*-promoter is not transcriptionally active in α -cells. In *ROSA^{mTmG};Pdx1-CreER* mice induced at the same time as the β ICKO cohort, we did not observe GFP related fluorescence in α -cells as shown in Suppl. Fig. 3E.

The changed section reads as follows:

ll.136 “To test for β -cell specificity of *Pdx1-CreER* driven recombination, we examined glucagon- and insulin-expressing islet cells in pancreatic sections from these reporter mice (Suppl

Fig. 3E). We did not observe recombination in glucagon-positive cells of these mice, indicating that at the time of induction, *Pdx1* is specific to β -cells.”

It might be of interest to the reviewer that we have indeed observed defects in glucagon secretion

that are potentially related to defects in cell-cell-communication via EphA/ EphrinA signalling or other pathways (see below). We are currently striving to better understand this phenotype but feel it would be beyond the scope of this study focusing on β -cells and insulin secretion.

2. The histograms shall be shown as dot plots, which enable the reader to identify possible subgroups that might otherwise be hidden.

Thank you for your suggestion. We have since replaced several graphs with dot blots to better identify individual data points. These are shown in Fig. 1A and F, Fig. 3A, Fig. 5H and I, and Fig. 6C. In addition, we replaced the bar graphs for WB quantification with dot blots.

3. Several Western blots are overexposed. Even though this looks nice, a more scientific way is to show non-saturated Western blots wherever possible

Thank you for raising this point that of course has important consequences for WB quantification. All of the Western Blots shown in this version of the manuscript are non-saturated. We now use

ChemStudio SA² for WB documentation. Saturated pixels will automatically be pseudo-coloured to indicate a problem. As stated before, we only quantify and show bands that are not saturated.

4. The authors shall describe in the M&M section how they obtained islets from humans with type 2 diabetes, as this is currently missing

We apologize for our omission. As is the case for the islets from normoglycemic donors, we have obtained islets from diabetic donors from the Alberta Diabetes Institute IsletCore. We have expanded the M&M section accordingly:

“...Human primary islets from normoglycemic and T2DM donors were supplied by the Alberta Diabetes Institute IsletCore (Edmonton, Canada), supported by the Alberta Diabetes Foundation (ADF), the Human Organ Procurement and Exchange (HOPE) and the Trillium Gift of Life Network (TGLN) for coordinating donor organs the Islet Core...”

Reviewer 2:

*Overall, this is a very interesting study reporting a novel signaling axis downstream of primary cilia in pancreatic β -cells. The biochemical evidence between *Ift88* and *EphA* receptors is convincing, but the downstream connection with the *Rac1*-actin cytoskeleton is a bit tenuous and sometimes confusing – this could be expanded or removed. Additional specific comments are listed below.*

Thank you for your comments. We have expanded on the connection with the actin cytoskeleton and hope we can convince you that *Ift88* plays a role in actin reorganization related to *Tiam1* and *Rac1* activity.

Specific comments:

1) The data obtained in vivo from the β ICKO animals and ex vivo KO islets are quite convincing, including GTT, insulin secretion and perfusion assays. Less convincing are the results obtained in the rescue experiments with recombinant EphrinA5-Fc chimera (Fig. 3) as well as with Tiam inhibitor NSC23766 (Fig. 5). The perfusion experiment shown in Fig. 1 shows a dramatic difference in insulin secretion between control and the β ICKO islets, which cannot be appreciated anymore in the rescue experiments. Why the authors have not always included the untreated control islets as baseline? This might help to appreciate the differences between control and mutant versus the rescued insulin secretion.

We apologize for the confusion, we did not show untreated control islets as baseline in an attempt to present the data as clearly as possible and to avoid showing the same data set twice. Obviously we did not succeed. As requested, we added both induced and control untreated β ICKO islets to the graphs in Fig. 3 A and 5H. We hope that the effects of EphrinA5-Fc and NSC23766 treatment respectively can be appreciated more readily now.

2) The differences in phosphorylation shown by WBs with the p-EphA3 antibody in sh- lft88 knockdown MIN6 line treated with EphrinA5-Fc or Ptp1b inhibitor are not convincing, given the images shown in Fig. 3D and Fig.4 G. Are these images/ gels overexposed? This is an important point in particular with regard to the EphrinA rescue.

Thank you for raising this point that of course has important implications for quantification and therefore interpretation of our results. As stated previously, we are showing only unsaturated WB exposures in this version of the manuscript. We use *ChemStudio SA²* for WB documentation;

saturated pixels will automatically be pseudocoloured to indicate a problem. We only quantify and show non-saturated bands.

3) Does the increased phosphorylation affect only EphA2 and EphA3 receptors or all EphA receptors? Why the authors characterize only EphA3 and nothing is shown about EphA2?

We truly regret that we are unable to answer whether or not the effect is specific to EphA2 and EphA3. The only specific antibodies to pTyr EphA are those blotted in the PathScan antibody arrays we used (Suppl. Fig. 4). Upon validation, pEphA2 antibody gave very weak bands and the EphA2 antibody did not work in immunoblotting, making validation of the PathScan result very difficult. The bigger effect was observed for EphA3, so we focused on this receptor. Again, we apologize for not being able to give a more definitive answer, but the limited availability of suitable reagents hinders more detailed investigations of these matters. The text was revised to reflect these difficulties:

ll.153: “...We **confirmed** EphA3 hyper-phosphorylation in Tx-treated β ICKO islets by immunoblotting (Fig. 2A...). **We were unable to validate excessive levels of pEphA2 due to a lack of suitable antibodies...**”

4) The IF analysis is important and should be expanded and improved. First, why in the IF staining the authors used an antibody against EphA5 instead of EphA2 or EphA 3 (Fig. 3F) ? See also point above, is the effect specific ? Second, what is shown in the IF images? Single cells or clusters? What do the dotted lines demarcate? Third, EphA5 staining seems increased in the sh-lft88, but the authors stated in the Result section that the effect is only on phosphorylation and not on total protein level. How do they explain this discrepancy?

The IF should be supported by quantitative analysis and co-localization between GM130 and EphA should be measured.

Again, we deeply regret not giving a more straightforward and definitive answer. We were unable to find an anti-EphA3 antibody suitable for immunofluorescent staining. Instead, we had used one raised against EphA5 as a surrogate. We realize that this is unsatisfactory and removed those sections. In the absence of suitable antibodies, we overexpressed *EphA3-myc* in control and *lft88*-depleted cells. We hope that we can convince you that differences in EphA3 localization are in good agreement with the other lines of evidence and reflect defects in EphA3 internalization when *lft88* is lost. This is shown in Fig. 4F. Before addition of EphrinA5-Fc, EphA3-myc is distributed throughout the cytoplasm in control cells. In *shlft88* stable cells, the pattern is less evenly distributed. In both cells, a subpopulation of EphA3-myc is presented at the plasma membrane, although it seems to be increased in *shlft88* compared to controls. After addition of EphrinA5-Fc, a promiscuous ligand with the highest binding affinity to EphA3, control cells internalize EphA3-myc efficiently with the majority of the receptor decorating the perinuclear region. By contrast, *lft88*-depleted cells show myc-specific immunofluorescence in the periphery, in proximity to the plasma membrane, consistent with a failure to efficiently internalize EphA3-EphrinA5-Fc.

The text was changed accordingly and now reads as follows:

ll. 282: "...None of the commercially available antibodies raised against EphA3 were suitable for immunofluorescent staining in our hands. Therefore, we used a myc-epitope tagged EphA3 expression plasmid, *EphA3-myc*, to visualize EphA3 localization (Fig. 4F). After stimulation with EphrinA5-Fc, EphA3-myc is efficiently internalized and predominantly found in the perinuclear region of control MIN6m9 cells. By contrast, a large subpopulation of EphA3-myc is observed in the periphery of *lft88*-depleted cells, proximal to the plasma membrane. These data

are in good agreement with the other lines of evidence and suggest that EphA3 internalization is dependent on Ift88 function...”

As suggested by the reviewer, we quantified the co-localization of Tiam1 with CGN marker GM130 by determining the Pearson correlation coefficient. The results are shown in Fig. 5I. In the panels, white lines demarcate the CGN as marked by GM130. Correlation analysis revealed a significant decreased in Pearson coefficient, indicative of a less tight spatial correlation of Tiam1 and GM130. The changed passage in the manuscript now reads as follows:

ll. 391: “...In addition to protein levels, we also tested the localization of Tiam1 in *shIft88* expressing MIN6m9 cells. We found Tiam1 immunofluorescent intensity widely distributed throughout the cytoplasm in Ift88 1 and 2 and seemingly strongly increased (Fig. 5I). In control cells, Tiam1 is associated with the GM130-positive cis-Golgi network (CGN). To better quantify the change in cellular localization, we determined Pearson’s correlation coefficient and observed a significant decrease in Ift88-depleted MIN6m9 derivative cell lines. This is indicative of a less strict spatial association between Tiam1 and the CGN compared to controls...”

5) The connection of Ift88 and Rac-actin cytoskeleton is not convincing. For ex. no data is included about the actin cytoskeleton organization or manipulation of actin polymerization.

We apologize for the confusion. We hope you will agree that we have made substantial advances in our understanding of the link between loss of *Ift88* function, Rac1/ Tiam1 and actin dynamics. The role for ciliary and basal body proteins such as Ift88 or Bbs4, Bbs6 and Bbs8 in actin reorganization has been previously reported. To better understand the nature of this role, we tested the ratio of actin monomers (G-actin) to filamentous actin (F-actin) and found a shift

towards higher levels of actin polymerization when MIN6m9 cells were depleted of *Ift88* protein (Fig. 5A). Similarly, phalloidin labelling of F-actin revealed morphological differences between control and *Ift88*-depleted cells (Fig. 5B). To address the role of actin polymerization in endocytosis, we treated MIN6m9 cells in presence or absence of *Ift88* with Cytochalasin D, an inhibitor of actin polymerization. We observed that EphrinA5-Fc internalization was ablated in all cells regardless of presence or absence of *Ift88* (Fig. 5C). It has been shown that one of the signalling pathways regulating actin dynamics is planar cell polarity signalling, part of non-canonical, β -catenin independent Wnt signalling. We and others observed that loss of cilia/ basal body integrity affects PCP signalling and is concomitant to stabilization of β -catenin (Gerdes et al., Nat Genet, 2007). Here, we found that β -catenin protein levels are upregulated in *Ift88*-depleted MIN6m9 cells (Fig. 5D). We also found that Tiam1, a GEF for Rac1, is upregulated on the mRNA and protein level (Fig. 5D and E). Tiam1 is also part of protein complex with Par3 and Par6 controlling cell polarity, indicating a polarity defect in *Ift88*-depleted cells. Because the shift in F- to G-actin is also observed in delaminating cells, we checked for markers of epithelial-mesenchymal transition and found Snail, Slug (Snail2) and Vimentin significantly upregulated in *Ift88*-depleted MIN6m9 cells, indicating that the cells are gaining more mesenchymal-like properties. At the same time, E-cadherin protein levels are decreased, suggesting that these cells lose epithelial-like characteristics (Fig. 5D). The figures have been changed accordingly and the manuscript has been expanded. We added a subsection "*Ift88* is involved in maintaining epithelial like polarity" (II.326). We hope that these lines of evidence can convince you that actin dynamics are changed in *Ift88*-depleted MIN6m9 cells, likely due to a shift in cell polarity, that cilia-regulated actin dynamics play a role in EphrinA5-Fc internalization and that inhibition of Tiam1 can partially reverse the inhibition of insulin secretion.

The writing of the manuscript could be improved. Both Introduction and discussion are long and wordy, the important messages do not come through and it is sometime difficult to appreciate the novelties compared to previous literature on cilia in beta-cells.

We apologize for the confusion and have significantly shortened and modified both introduction and discussion of the manuscript to more clearly state our message. In addition, the manuscript was prepared after consulting a professional writer and native speaker to improve the writing in general. We hope that these measures have helped to make our point more clearly.

Reviewer 3:

These findings are novel and interesting in that they has firstly demonstrated a functional link between primary cilia and Eph-ephrin signaling and abnormal Eph signaling in human pancreas from diabetic subjects. However, they were considerably limited by a lack of control and statistis, small n numbers, and lots of errors in the manuscript and figures.

Major comments.

1. To exclude any possible effects of tamoxifen on beta-cells and insulin secretion, another

control group (Pdx-Cre negative, and tamoxifen-injected) was needed in Figure 1.

Thank you for your comment. We have used both vehicle-treated β ICKO mice as a

control for Tamoxifen treatment and Tamoxifen treated *Ift88*^{loxP/loxP} mice to control for effects of Cre-expression as mentioned in the text:

ll.91: "...To control for effects of Tx-treatment and *Pdx1-CreER* overexpression, both vehicle-treated β ICKO mice and Tx-treated *Ift88*^{loxP/loxP} mice from the starter strain served as controls..."

To avoid overcrowding of the figures, we did not show both controls in the figures. We never observed significant differences in glucose tolerance or insulin secretion between vehicle-treated β ICKO mice and Tamoxifen-treated *Ift88*^{loxP/loxP} mice. Above, we show GTT excursion curves at eight weeks to demonstrate there is no significant difference between the two control groups.

2. The n numbers of dynamic insulin secretion experiments in Fig 1F, 3A, and 5D were only 2. How many times were these experiments repeated to conclude the findings?

The dynamic insulin secretion experiment were repeated 2 times using islets pooled from 5 different animals (N=5, two independent experiments, technical duplicates). Pooling animals should dilute differences between individuals. To increase the statistical power and therefore exclude false negative findings, we repeated the experiment 2 additional times for a total of n=4, islets pooled from 10 animals.

3. How does ciliary dysgenesis lead to increased Tiam-1 expression? The molecular link mediating this changes needs to be suggested or discussed.

We deeply regret that technical limitations make it impossible to give a direct answer, f.e. via reverse ChIP that has not been established yet. Instead, we did some literature research and

found that Tiam1 and GTP-bound Rac1 enhance β -catenin/ TCF-dependent transcription in certain types of colorectal cancer. We have previously shown a link between primary cilia and non-canonical Wnt/ planar cell polarity signaling. Tiam1 forms a super-complex with the polarity complex Par/Par6/aPKC. It is possible that Tiam1 upregulation is an attempt to compensate for loss of polarity signaling. To support our hypothesis, we tested and found an increase in β -catenin that we have previously shown to be concomitant to the loss of PCP signaling (Fig. 5). β -catenin dependent Wnt signaling is one of the known inducers of Epithelial-mesenchymal transition (EMT) and we found an increase of mesenchymal markers Snail, Slug, and Vimentin, and a decrease in epithelial marker E-cadherin (Fig. 5D). Taken together, we suggest that loss or rather a shift in cell polarity leads to an increase in Tiam1 and affects actin dynamics.

4. The authors showed increased EphA phosphorylation in islets from the subjects with diabetes. However, they did not show ciliary changes in these subjects. Is EphA hyperphosphorylation linked to ciliary changes in human islets?

Unfortunately that limited availability of tissues from diabetic organ donors makes it very difficult to answer this question with sufficient statistical power. Clearly, further investigations are warranted and we are in the process of contacting consortia that have collected data from larger patient cohorts. Because these studies are very time consuming, we regret that the direct answer will be beyond the scope of this manuscript. However, we were able to obtain access to Affymetrix gene expression data from normoglycemic, pre-diabetic and diabetic individuals that underwent partial pancreatectomies at the University of Tübingen. Islets were isolated by laser capture microdissection and differential gene expression was analyzed (Table1). We found that RAB3IP, a direct interaction partner of the BBSome is significantly downregulated in diabetic patients and negatively correlates with HbA1c glycosylation and fasting glucose. It correlates with insulin secretion. This provides a direct link between cilia function, vesicle transport and Type 2 diabetes.

We have added a table and amended the Results and Discussion sections accordingly. They are now worded as follows:

ll. 435 (Results) “...To test for a more direct link between islet cilia/ basal body function and Type 2 Diabetes, we analyzed islet transcriptomes obtained from pancreatic resections of human donors. In a cohort (N=19) of non-diabetic, prediabetic and diabetic individuals⁴⁷, we found significant misregulation of *RAB3IP* (*RAB3A interacting protein*), a direct interaction partner of BBSome component BBS1 and a GEF of RAB8A.⁴⁸ *RAB3IP* mRNA levels associate negatively with increases in fasting glucose and positively with insulin secretion, i.e. HOMA2%B⁴⁹ calculated using C-peptide and insulin (Table 1). This finding supports a functional role for cilia/ basal bodies in insulin secretion and islet function in humans...”

ll.493 (Discussion) “...Finally, several lines of evidence support a role for cilia/ basal body function in insulin secretion in humans: When we depleted islets from four different donors of *IFT88*, we observed increased EphA3 phosphorylation levels and impaired glucose-stimulated insulin secretion, both in good agreement with our findings in Tx-treated β ICKO mice. We observed increased pEPHA3 levels in islets from diabetic donors compared to normoglycemic controls. And misregulation of *RAB3IP* in islets from diabetic donors supports a direct link between cilia/ basal body dysfunction, vesicle trafficking and Type 2 *Diabetes mellitus*. Taken together, we propose that modulation of ciliary function and endosomal signal processing could offer a novel approach to therapeutic interventions in metabolic diseases such as Type 2 *Diabetes mellitus*...”

We also took care to address the minor comments you had.

In conclusion, we hope we have been able to address the comments of all reviewers to their satisfaction.

Reviewer #2 (Remarks to the Author):

In the revised version of this manuscript the authors have satisfactorily responded to all the points raised by this referee.

The new set of experiments included has greatly improved the work.

Reviewer #3 (Remarks to the Author):

The authors well adressed the questions. The revised manuscript is suitable for the publication in Nature Communications.

1. Please include the data of OGTT in two different control groups as a form of supplemental information.
2. Please discuss the necessity to investigate β -cell ciliary changes in diabetic subjects in the future because the negative correlations of islet RAB3IP mRNA expression and fasting blood glucose levels or HOMA-beta index provided a very indirect evidence for ciliary signaling and insulin secretion in human islets.

Again, we wish to thank all reviewers for their time and efforts in improving our manuscript. We hope our revisions will meet with your approval. Changes in the manuscript from the previously submitted version are highlighted in yellow.

Reviewer #1 (Remarks to the Author):

I looked at the authors' rebuttal to all three referees' comments and found that the authors extensively revised and addressed the points of critique and suggestions, except for the minor points raised by referee 3 that are not addressed at all in the rebuttal.

We apologize for the confusion. We had addressed all minor comments without detailing the remedies in response to reviewer #3. To clarify, we list the changes below:

Reviewer #3: Minor comments

1. In Fig 2a, total EphA blot was missing. Fig 2A legend indicated phosphorylated p44/42 MAPK but p44/42 MAPK blot were also missing.

We corrected the figure legend, p44/42 MAPK is displayed in Fig 2C, not 2A. We normalized pEphA3 to γ -tubulin, not EphA.

2. Fig 2B-D demonstrated the successful knockdown of IFT88 with shRNA transfection. These data should be presented as a Supplementary data.

Thank you for your suggestion, we have now moved the data from Fig 2B-D to Suppl Fig 1D-F

3. In Fig 2F the first blot, the protein loading control i.e. pan Erk blot is needed.

We apologize for our omission and have added the panErk blot to Fig 2C (previously Fig 2F)

4. Fig 3A and 5B had three groups. So AUC graphs should have three groups. Both tamoxifen-untreated and ephrinA5-Fc or NSC23766-untreated control is needed.

We thank you for your suggestion, we modified Figs 3A and 5B and added insulin secreted from untreated and tamoxifen-treated controls along with that from stimulated islets, both in the line graph and in the AUC graph.

5. Fig. 3C, x axis should have 4 groups. Moreover, data of low and high glucose condition should be shown in rectangles but there were only circles and triangles in the figure.

We apologize for the confusion. We have now changed the graph symbols to more easily identify the different groups. Now all the symbols are squares (blue and red) for the different glucose conditions.

6. Incomplete sentence in page 9 line 123-125.

The sentence in page 9 line 123-125 has been rewritten and now reads as follows:

“...Islets from Tx-treated β ICKO mice 20 weeks post induction had significantly reduced insulin secretion after incubation with 11 mM glucose for 30 min (Suppl. Fig. 3A n=5, Vehicle: 7.1 fold increase \pm 1.2; Tx: 4.6 fold increase \pm 0.5; p=0.0025)...”

7. Page 10, line 142, *Primary cilia are not just involved in insulin signaling. Is this description true? The previous paper (ref 5) has shown impaired insulin signaling by ciliary disruption.*

Thank you for your comment. Indeed, a previous study from our lab showed an involvement of primary cilia in insulin signaling. In addition to insulin signaling, however, primary cilia are implicated in non-canonical Wnt signaling, Hedgehog signaling and transforming growth factor β (TGF β) signaling, to name only a few signaling pathways. In this sense, we stand by our statement.

8. *In Fig 3C, 3D, and 3E, the description in the text and figures are mismatched.*

9. *The description about Fig. 3B was ahead that of Fig.3A in the text.*

We apologize for the confusion, we corrected the text accordingly and changed the order in which Fig 3B and 3A are presented (line 183-195).

10. *Fig. 4F is missing in the text.*

The previous figure 4F (now 4G) has been added to the text and now reads as follows:

1292 ff

“...A Ptp1b specific inhibitor also abolished the different pEphA3 levels in control and *Ift88*-depleted MIN6m9 cells (Fig. 4G)...”

11. *Control without treatment with PTP inhibitor is needed.*

We have included no treatment controls in Fig 3D and Suppl. Fig. 5A. To avoid displaying the same data in several figures, we did not include this data in Suppl. Fig. 5B.

12. *In Fig. 5 A-C, there is no statistics in the figures. Some groups had no error bars.*

Error bars have been added to Fig 5E (Previously 5C) and dot blots replace the other bar graphs, making error bars obsolete.

13. *Fig 1A, repeated ANOVA rather than t-test should be performed to compare two groups each time point.*

Thank you for your comment. We have re-analyzed the data using repeated measure ANOVA ANOVA; in all cases, our findings remain statistically significant. The P values are stated in the text.

14. *In Fig. 3F, 4F and 5F, quantification is needed.*

We removed Figure 3F. Ptp1b-FLAG localizes to the perinuclear region as well as vesicles and plasma membrane. Because it is overexpressed, quantification of endogenous protein is not possible (Figure 4G, previously 4F). In Figure 5F, we added the Pearson coefficient for co-localization of cis-Golgi marker GM130 and Tiam1.

Only few remaining points remain to be addressed according to this referee:

1. *The authors argue that saturated western blot bands are pseudo-colored. That means to me that some of the bands shown in the Figures (e.g., γ -tubulin in Fig. 2A or ERK in Fig. 2C) must be colored, but this is not the case. Even though most papers in the past showed over-saturated blots,*

this is no longer state-of-the-art. Therefore, either the authors critically reflect on this issue in the main text to raise some caution on some of the blots, or the authors shall color the over-saturated bands in the main Figures.

First of all, we would like to thank the reviewer for raising an important point about WB in publications. We agree that quantification of WB bands is not trivial, not only because loading controls (and other bands) are sometimes oversaturated. Because amplification of the signal by antibody and secondary antibody binding is not linear, WB is semi-quantitative. To be as precise as possible, we use ImageStudioLite for quantification. Saturated pixels (Intensity=255) are pseudocoloured in white or blue (see below) and “infinity” is stated as the total luminescence intensity. γ -tubulin as loading control (Fig. 2A) is shown in red (see below), the infinity label refers to the green channel. Similarly, ERK (Fig. 2C) is not saturated. For comparison, we include here the raw image of the pERK/MAPK blot to show pseudocoloured pixels (arrows). Given the non-linearity of WB analysis, we base our conclusions on clearly visible differences in WB intensity and avoid relying on quantification exclusively.

Figure 1: screenshot of the raw blot in Fig 2A, γ -Tubulin is pictured in red.

Figure 2: screenshot of the raw blot in Fig 2C, panERK is quantifiable and no pseudocolors are present in the band (Ref figure 3 for example)

Figure 3: screenshot of the raw blot in Fig 2C, phosphoERK is quantifiable. Arrows indicate pseudocolor of over-exposed pixels as reference.

2. The glucose tolerance test with tamoxifen controls is useful and shows that this agent does not significantly alter glucose tolerance in mice. However, it would be good to incorporate this Figure somewhere in the actual manuscript and not just in the rebuttal, e.g. in a Supplementary Figure.

OGTT with two different control groups is now in Suppl Fig 2F, in the text the following paragraph was added (lines 90-93):

“...To control for possible side effects of Tamoxifen treatment or *Pdx1-CreER* expression, we included two different control groups, β ICKO mice treated with oil and *Ift88*^{loxP/loxP} mice treated with tamoxifen. Glucose tolerance was not affected in both animals and there were no statistically significant difference between the two controls groups (Suppl Fig 2F)...”

3. The Affymetrix data on *RAB3IP* expression in human islets from patients with diabetes, prediabetes or no diabetes are useful and support the findings, but must be better presented. Normally such data are confirmed using qPCR. Thus, it would be good to reproduce the Affymetrix data by qPCR with RNA from these human islets. As it is only one gene (*RAB3IP*), this should be possible with minute amounts of RNA. In addition, more information on the islet donors must be included in the Table legend, e.g. how many people were diabetic, prediabetic or non-diabetic.

We thank you for your suggestion and absolutely agree that this finding warrants further confirmation. Unfortunately, however, LCM islets from pancreas samples of partially pancreatectomized patients yield very little RNA and there is simply none left, even for qPCR confirmation of reduced *RAB3IP* expression. We have changed the manuscript accordingly to emphasize the need for a more detailed investigation of the role ciliary and basal body proteins play in the regulation of glucose homeostasis, insulin secretion and T2DM etiology.

We have removed the reference to *RAB3IP* expression from the abstract. The new sentence reads as follows:

ll 12: “...Defects in insulin secretion from IFT88-depleted human islets and elevated pEPHA3 in islets from diabetic donors both point to a role for cilia/ basal body proteins in human glucose homeostasis...”

ll 423: “...Although this finding tentatively supports a functional role for cilia/ basal bodies in insulin secretion and islet function in humans, we have to emphasize the need for further validation of our findings. Due to a lack of patient RNA samples, we could not confirm the expression data by

qPCR or another independent line of evidence. However, we suggest that our findings strongly support the need for a more detailed investigation of the role primary cilia and basal body genes and proteins might play in glucose homeostasis, insulin secretion and T2DM...”

ll. 492: “...And misregulation of *RAB3IP* tentatively supports a direct linke between cilia/ basal body dysfunction, vesicle trafficking and Type 2 *Diabetes mellitus*; still, further studies are warranted to more firmly establish such a link...”

Reviewer #2 (Remarks to the Author):

In the revised version of this manuscript the authors have satisfactorily responded to all the points raised by this referee.

The new set of experiments included has greatly improved the work.

We thank you for your comments that helped us improve this study.

Reviewer #3 (Remarks to the Author):

1. Please include the data of OGTT in two diffrent control groups as a form of supplemental information.

Thank you for your suggestion, we have added the GTT data to Supplemental Figure 2 (panel F) and changed the manuscript accordingly (l. 90)

“...To control for possible side effects of Tamoxifen treatment or *Pdx1-CreER* expression, we included two different control groups, β ICKO mice treated with corn oil and *Ift88*^{loxP/loxP} mice treated with tamoxifen. Glucose tolerance was not affected in both animals and there were no statistically significant difference between the two controls groups (Suppl Fig 2F)...”

*2. Please discuss the necessity to investigate β -cell ciliary changes in diabetic subjects in the future because the negative correlations of islet *RAB3IP* mRNA expression and fasting blood glucose levels or HOMA-beta index provided a very indirect evidence for ciliary signaling and insulin secretion in human islets.*

Thank you for your suggestion, we have edited the manuscript accordingly. For details regarding the edited manuscript, please refer to our response to Reviewer #1. We wholeheartedly agree that further studies are warranted to characterize potential β -cell ciliary changes in diabetic patients.

Reviewer #1 (Remarks to the Author):

The authors addressed the referees' remaining issues. However, they still do not include information on the human islet donors in Table 1, even though this had been requested (at least in the version I looked at). Information about diabetic, prediabetic and non diabetic subjects must be explicitly provided in the legend of Table 1 - it is not state of the art to not include this important information. If the authors do not include detailed information about the different islet groups they compare with each other, are the differences observed between e.g., the diabetic versus non diabetic islets just due to differences in gender? Or age? Meaning the authors report on confounder effects? Either they include this information or discuss in the text about these issues.

Point-by-point response to Reviewer Comment

Reviewer #1 (Remarks to the Author):

The authors addressed the referees' remaining issues. However, they still do not include information on the human islet donors in Table 1, even though this had been requested (at least in the version I looked at). Information about diabetic, prediabetic and non diabetic subjects must be explicitly provided in the legend of Table 1 - it is not state of the art to not include this important information. If the authors do not include detailed information about the different islet groups they compare with each other, are the differences observed between e.g., the diabetic versus non diabetic islets just due to differences in gender? Or age? Meaning the authors report on confounder effects? Either they include this information or discuss in the text about these issues.

We profusely apologize for the confusion, the current as well as the previous version of the manuscript contains a supplementary table (Suppl. Table 1) containing all the relevant data regarding clinical parameter of the cohort.